# Dynamic-gravimetric preparation of metrologically traceable primary calibration standards for halogenated greenhouse gases

**Myriam Guillevic**[1], **Martin K. Vollmer**[2], **Simon A. Wyss**[2], **Daiana Leuenberger**[1,3], **Andreas Ackermann**[1], **Céline Pascale**[1], **Bernhard Niederhauser**[1], and **Stefan Reimann**[2]

[1]Federal Institute of Metrology METAS, CH-3003 Berne, Switzerland
[2]Laboratory for Air Pollution and Environmental Technology, Empa, Swiss Federal Laboratories for Materials Science and Technology, Dübendorf, Switzerland
[3]Now at: Federal Office for the Environment FOEN, Air Pollution and Chemicals Division, CH-3003 Berne, Switzerland

*Correspondence to:* M. Guillevic (myriam.guillevic@metas.ch)

**Abstract.** For many years, the comparability of measurements obtained with various instruments within a global-scale air quality monitoring network has been ensured by anchoring all results to a unique suite of reference gas mixtures, also called 'primary calibration scale'. Such suites of reference gas mixtures are usually prepared and then stored over decades in pressurised cylinders by a designated laboratory. For the halogenated gases which have been measured over the last forty years, this anchoring method is highly relevant as measurement reproducibility is currently much better ($< 1\%$, $k = 2$ or 95 % confidence interval) than the expanded uncertainty of a reference gas mixture (usually $> 2\%$). Meanwhile, newly emitted halogenated gases are already measured in the atmosphere at sub-pmol/mol levels, while still lacking an established reference standard. For compounds prone to adsorption on material surfaces, it is difficult to evaluate mixture stability and thus variations in the molar fractions over time in cylinders at pmol/mol levels.

To support atmospheric monitoring of halogenated gases, we create new primary calibration scales for $SF_6$ (sulfur hexafluoride), HFC-125 (pentafluoroethane), HFO-1234yf (or HFC-1234yf, 2,3,3,3-tetrafluoroprop-1-ene), HCFC-132b (1,2-dichloro-1,1-difluoroethane) and CFC-13 (chlorotrifluoromethane). The preparation method, newly applied to halocarbons, is dynamic and gravimetric: it is based on the permeation principle followed by dynamic dilution and cryo-filling of the mixture in cylinders. The obtained METAS-2017 primary calibration scales are made of 11 cylinders containing these five substances at near ambient and slightly varying molar fractions. Each prepared molar fraction is traceable to the realisation of SI units (Système International d'Unités) and is assigned an uncertainty estimate following international guidelines (JCGM 100:2008), ranging from 0.6 % for $SF_6$ to 1.3 % ($k = 2$) for all other substances. The smallest uncertainty obtained for $SF_6$ is mostly explained by the high substance purity level in the permeator as well as low $SF_6$ contamination of the matrix gas. The measured internal consistency of the suite ranges from 0.23 % for $SF_6$ to 1.1 % for HFO-1234yf ($k=1$). The expanded uncertainty after verification (i.e. measurement of the cylinders vs each others) ranges from 1 % to 2 % ($k = 2$).

This work combines the advantages of SI-traceable reference gas mixture preparation with a calibration scale system for its use as anchor by a monitoring network. Such a combined system supports maximising compatibility within the network while linking all reference values to the SI and assigning carefully estimated uncertainties.

For $SF_6$, comparison of the METAS-2017 calibration scale with the scale prepared by SIO (Scripps Institution of Oceanography, SIO-05) shows excellent concordance, the ratio METAS-2017/SIO-05 being 1.002. For HFC-125, the METAS-2017 calibration scale is measured as 7 % lower than SIO-14, and for HFO-1234yf 9 % lower than Empa-2013. No other scale for HCFC-132b was available for comparison. Finally, for CFC-13 the METAS-2017 primary calibration scale is 5% higher than the interim calibration scale (Interim-98) that was in use within the Advanced Global Atmospheric Gases Experiment (AGAGE) network, before adopting the scale established in the present work.

# 1 Introduction

Since the 1970s, atmospheric measurements of CFCs (chlorofluorocarbons) and HCFCs (hydrochlorofluorocarbons), used as refrigerants and blowing agents, have evidenced their role in stratospheric ozone layer depletion (Molina and Rowland, 1974; WMO, 1981). The reduction of CFC use (and later HCFCs) has been under strict regulations of the Montreal Protocol on Substances that Deplete the Ozone Layer since its entering into force in 1989 (WMO, 2014). While the molar fractions of major CFCs are now declining in the atmosphere, some longer-lived minor CFCs are still increasing (WMO, 2014; Laube et al., 2014; Vollmer et al., 2018). HFCs (hydrofluorocarbons) were introduced as replacement for CFCs and HCFCs. Their emissions, though not harmful to the ozone layer, are still increasing and contributing to global warming due to their high radiative forcing (Harris and Wuebbles, 2014; Velders et al., 2009). For this reason the recent Kigali Amendment (Oct. 2016) added these HFCs to the Montreal Protocol. As a consequence and in compliance with European Directive 2006/40/EC, replacement compounds were introduced, foremost the HFOs (hydrofluoroolefins, Vollmer et al., 2015).

Non-Article 5 (developed) countries of the Montreal Protocol are bound to report their CFCs/HCFCs/HFCs production and consumption based on so called 'bottom-up' inventories. Within the Kyoto Protocol aiming at limiting climate change, Annex-1 (developed) countries report inventories of greenhouse gas emissions to the atmosphere, for which reduction targets are set. Reviews of the success of such reduction targets and projections of future developments are so far still based on those inventories, while ozone layer recovery and climate change depend on real atmospheric molar fractions.

Atmospheric measurements of halogenated compounds are currently provided by several networks such as AGAGE (Advanced Global Atmospheric Gases Experiment), NOAA (National Oceanic and Atmospheric Administration), and GAW (Global Atmospheric Watch). Such measurements, used to precisely estimate atmospheric molar fraction of these halogenated substance together with associated trends, are crucial to understand and predict the evolution of stratospheric ozone and estimate their radiative forcing thereby refining future climatic projections. Furthermore based on these measurements and using atmospheric transport modeling, emissions can be quantified ('top-down' estimation, e.g., Prinn et al., 2000; Rigby et al., 2010; Brunner et al., 2017). The comparison of top-down reconstructions with bottom-up inventories shows agreement for some gases but also discrepancies that can be considerable for others (Weiss and Prinn, 2011; Lunt et al., 2015; Hu et al., 2016; Simmonds et al., 2016; Sherry et al., 2017). The top-down approach thus is a complementing and independent way to review production/consumption/emission inventories and compliance with reduction targets, while assessments of climate

forcing and stratospheric ozone rely on observations of atmospheric composition.

To quantify atmospheric molar fractions at specific sites, detect gradients between monitoring stations, evaluate data consistency and lack of biases and thereby attribute emissions to specific geographical areas, results have to be linked to accurate calibration scales. Currently for most monitored halogenated gases, measurement precision is as low as 0.4 % ($k = 2$ or 95 % confidence interval, Miller et al., 2008). This is much better than the expanded uncertainty of reference gas mixtures, estimated to be in the order of e.g. 2 to 4 % for SIO standards (Scripps Institution of Oceanography, e.g., Prinn et al., 2000), 0.3 to 3 % for NIST (National Institute of Standard and Technology, Rhoderick et al., 2015), and 0.6 to <2 % for NOAA (Hall et al., 2007; Montzka et al., 2015; Lim et al., 2017). Given this technical state of the art, monitoring networks have developed a so called 'primary calibration scale' system, in which all stations of a specific network are anchored to the same suite of primary reference gas mixtures with a calibration chain as short as possible. To ensure long term continuity of atmospheric composition data, such calibration scales have to be maintained over years and only replaced if substantial necessity arises, with a scheme to properly back-calibrate all past measurements if a new calibration scale is defined. Accurate and stable calibration scales directly impact the quality of emission quantification, trend estimation and the calculation of values relevant for future climate projections, such as radiative forcing which is derived from atmospheric composition.

The calibration scale approach enables a high degree of consistency, but still requires detecting and documenting systematic offsets as an indicator for potentially existing systematic biases. This QA/QC procedure includes regular intercomparisons between instruments installed at the same sites and reference gas mixtures or air sample flask exchanges (Hall et al., 2014; Rhoderick et al., 2015). In addition, geographically close monitoring stations anchored to different calibration scales regularly assess the potential development of a bias over time, that can dependent on the molar fraction (e.g., O'Doherty et al., 2004; Rigby et al., 2010; Vollmer et al., 2016; Simmonds et al., 2017).

The production of a robust and accurate primary calibration scale at low molar fraction level, which is usually created through a set of reference gas mixtures, is a major task (Prinn et al., 2000; Zhao et al., 1997; Dlugokencky et al., 2005; Zhao and Tans, 2006; Hall et al., 2007). In particular for compounds prone to adsorption on material surfaces, it is difficult to evaluate mixture stability and thus variations of molar fractions over time in cylinders. As a consequence, some halogenated gases are measured in the atmosphere against reference standards that lack conventional calibration (relative calibration scales) or with only limited sets of primary reference standards (e.g., Vollmer et al., 2015).

To support atmospheric monitoring of halogenated gases, we present here a method to produce reference gas mix-

tures at near atmospheric molar fractions for $SF_6$ (sulfur hexafluoride), HFC-125 (pentafluoroethane), HFO-1234yf (HFC-1234yf, 2,3,3,3-tetrafluoroprop-1-ene, newly emitted compound), HCFC-132b (1,2-dichloro-1,1-difluoroethane)

[5] and CFC-13 (chlorotrifluoromethane). While $SF_6$ and HFC-125 have already widely used calibration scales (see Sect. 4.1), HCFC-132b and CFC-13 have been measured for many years in AGAGE yet not reported on a conventional calibration scale (Vollmer et al., 2018). We apply a prepara-

[10] tion technique combining dynamic gravimetry (ISO 6145-10, 2002) and dynamic dilution (ISO 6145-7, 2009), followed by cryo-filling in cylinders. This is an alternative to the prevailing preparation method for such compounds which is by static gravimetry (Prinn et al., 2000; Hall et al., 2007;

[15] Rhoderick et al., 2015; Lim et al., 2017). The produced suite of reference gas mixtures is SI-traceable, i.e. all measured/relevant quantities of each preparation step are linked to the realisation of SI units (the International System of units) through an unbroken chain of calibrations. Finally,

[20] the prepared suite of mixtures is assigned a carefully quantified expanded uncertainty taking into account all known relevant potential sources of uncertainties, following international recommendations (JCGM, 2008). The aim is to provide independent calibration scales and compare them to

[25] other available scales. This would provide existing calibration scales, which have so far been used on a relative basis, with a link to the SI.

In this paper, we present in detail the method developed to prepare SI-traceable reference gas mixtures for the men-

[30] tioned halogenated gases at near atmospheric background levels, i.e. as low as 1 pmol/mol (Sect. 2). The calculations to assign prepared values are described together with the method. The associated uncertainty budgets, established following JCGM:2008, are presented and discussed in Sect. 2.5.

[35] The internal consistencies of the calibration scales are determined in Sect. 3.2. These SI-traceable primary standards are then compared to other standards currently in use in the AGAGE network, as reported in Sect. 4. In addition, we determine conversion factors which compare the METAS-2017

[40] suite to other primary calibration scales.

## 2 Method

### 2.1 Method overview: Dynamic-gravimetric generation process

The suite is designed to consist of one master cylinder (here-

[45] after MP-001, for METAS Primary cylinder n. 001) containing all components at near ambient molar fractions but none below 1 pmol/mol, in order to not exceed preparation uncertainties of 2 %. Ten additional cylinders are filled with molar fractions bracketing those filled in cylinder MP-001 over the

[50] range from 20 % less to 30 % more. The resulting prepared molar fraction range covered by this suite varies between

the five compounds, with a range of 0.9 – 1.5 pmol/mol for HFO-1234yf with the lowest molar fractions, to 26 – 42 pmol/mol for HFC-125 with the highest molar fractions (see details for each substance in Table 3). This allows the later [55] determination of the internal consistency of the suite.

The generation process consists of three successive steps, starting from pure halocarbon substances diluted to molar fractions as low as pmol/mol in synthetic air. In a first step a matrix gas is spiked with one pure halocarbon substance us- [60] ing a permeation device (Sect. 2.2). In a second step, the high molar fraction mixture is dynamically diluted to pmol/mol level mixture using thermal mass flow controllers (MFCs, Sect. 2.3). In the final step the mixture is successively transferred into the 11 cylinders by cryo-filling (Sect. 2.4). In or- [65] der to generate multi-component mixtures, the permeation device is changed and all steps are repeated with the mixture containing a new substance being added to the same suite of cylinders.

### 2.2 Permeation [70]

Reference gas generation by applying the permeation method combined with dynamic dilution is an established, standardised technique, particularly for reactive gases (e.g., O'Keeffe and Ortman, 1966; Scaringelli et al., 1970; Brewer et al., 2011; Flores et al., 2012; Haerri et al., 2017). It is routinely [75] used at METAS following a procedure in compliance with international standards (ISO 6145-10, 2002; ISO 6145-7, 2009). The permeation method is based on constant transfer of the substance of interest from a permeation device (or permeator), resulting in a mass loss which can be continuously [80] monitored. Permeation devices are placed in a permeation chamber, i.e. a controlled atmosphere in terms of temperature and pressure, continuously flushed by a carrier gas stream. Permeators used here consist of a stainless steel reservoir containing the pure substance as a liquid, sealed with a cap [85] containing a polymer membrane permeable for the specific substance. All permeators used in this study (Fine Metrology, Italy) were filled with substances of purity 99 % or higher (Synquest Laboratories, Florida, USA). Substance purity levels were determined by the manufacturer using flame ioni- [90] sation detector analysis. Permeators are filled under laminar flow to avoid potential contamination.

The permeation rate depends exponentially on temperature; secondary influences are carrier gas composition and pressure (Lucero, 1971; Moosbach and Hartkamp, 1993; [95] Jost, 2004; Brito and Zahn, 2011; Haerri et al., 2017). For mass loss determination, each permeator is placed individually in our magnetic suspension balance (MSB) 'Violetta' (hereafter MSB-Violetta, model FLUIDIFF MP, installed in 2014, Rubotherm, Germany, Fig. 1). A MSB system allows [100] for continuous and unperturbed mass measurements as the temperature-controlled chamber, where the permeator is suspended, is physically decoupled from the balance itself. The stainless-steel permeation chamber of MSB-Violetta allows

**Table 1.** Overview of cylinders used to store the reference gas mixtures.

| Cylinder reference | MP-003 to MP-006 | MP-001 | MP-002 | MP-007 to MP-011 |
|---|---|---|---|---|
| Manufacturer | Swagelok | Essex Industries, MO, USA | | |
| Material | Stainless steel 304L | Stainless steel 304L | | |
| Treatment | SilcoNert2000 coating | Electropolishing | | |
| Volume | 2.25 L | 34 L | 34 L | 4.5 L |
| Pressure | 60 bar | 65 bar | 30 bar | 20 bar |
| Volume filled, L @ STP | 135 | 2210 | 1020 | 90 |
| Surface/volume, m$^{-1}$ | 1 | 0.4 | 0.8 | 2.5 |

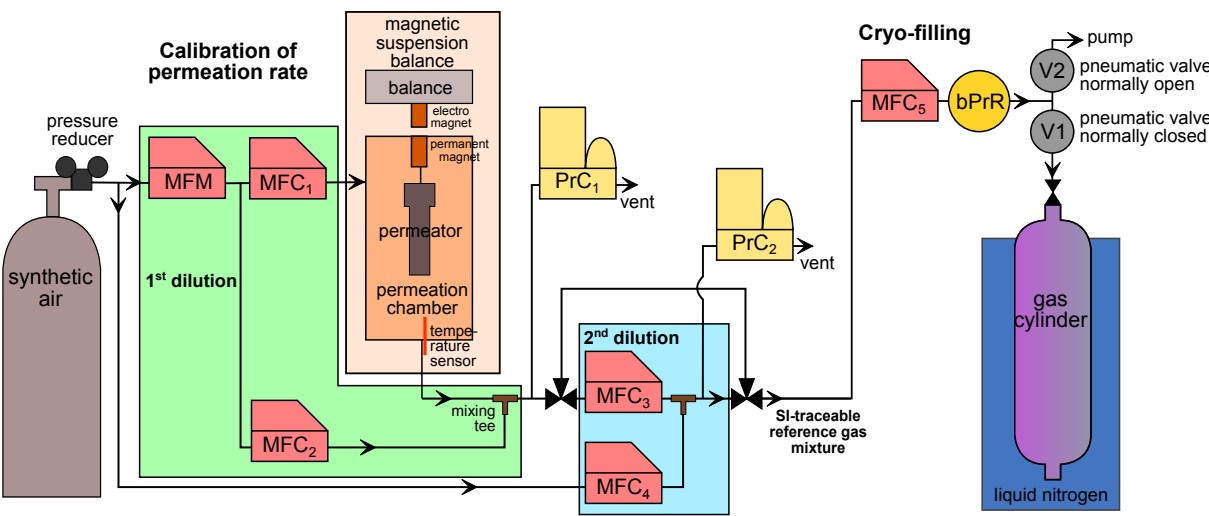

**Figure 1.** Schematic of the dynamic-gravimetric preparation method. MFM: thermal mass flow meter. MFC: thermal mass flow controller. PrC: pressure controller. bPrR: back (downstream) pressure regulator. V1 and V2: pneumatic valves.

for precisely controlling temperature (± 0.02°C, measured with a Pt100 sensor), pressure (± 0.1 hPa, Bronkhorst El-Press, the Netherlands) and carrier gas flow (± 0.1 % of the flow, Red-y series, Vögtlin, Switzerland). In this study, per-
5 meation typically occurs at 45°C and 3500 hPa. Inside the chamber, the permeator is coupled to a permanent magnet also placed in the chamber. On the outside of the chamber, an electromagnet connected to the balance (Sartorius ME66S) exerts a force over the permanent magnet thus coupling the
10 permeation unit to the balance. To minimise balance noise, total weight and position of the permeation device are adjusted, as well as the vertical position of the electromagnet. The absolute mass of the permeation unit is measured in three-minute intervals. To correct for balance drift, after ev-
15 ery 3 measurement points a calibration mass (CM1) is automatically placed on the balance plate. Additionally, a second calibration mass (CM2) of same volume but different mass is measured every 6 calibration points, to correct for any potential buoyancy change affecting the measurement of CM1.
20 The masses of CM1 and CM2 are traceable to the Swiss realisation of the kilogramm (Fuchs et al., 2012; Marti et al., 2015; Marti, 2017, and references therein). All weight measurements and associated corrections realised with MSB-

Violetta are fully automated. After each opening and closing of the permeation chamber, its tightness is checked by clos- 25 ing the input gas from the synthetic air cylinder as well as all exits and checking the absence of pressure decrease over time.

After inserting a permeation device, a stabilisation period is required mainly depending on chamber temperature, pres- 30 sure and permeator membrane properties, before the mass loss becomes linear over time. This linear mass loss vs time is then determined for at least 8000 min to minimise the standard deviation of the measured mass loss due to balance noise. The time window $t_{2,i} - t_{1,i}$ during which the mass data 35 are used is determined so that the residuals of the fit to the mass loss over time are centered around zero and randomly distributed (see example for CFC-13 on Fig. 2 and Section S1 in the Supplement). For each substance $i$, the molar fraction $x_{perm,i}$ of the mixture exiting the permeation chamber can 40 be calculated as:

$$x_{perm,i} = \frac{m_{1,i} - m_{2,i}}{t_{2,i} - t_{1,i}} \cdot purity_i \cdot \frac{1}{M_i} \cdot \frac{V_{m,carrier}}{q_{V1,i}} + x_{res,i} \quad (1)$$

with $m_{1,i} - m_{2,i}$, mass difference in between beginning and end of linear mass loss (g);

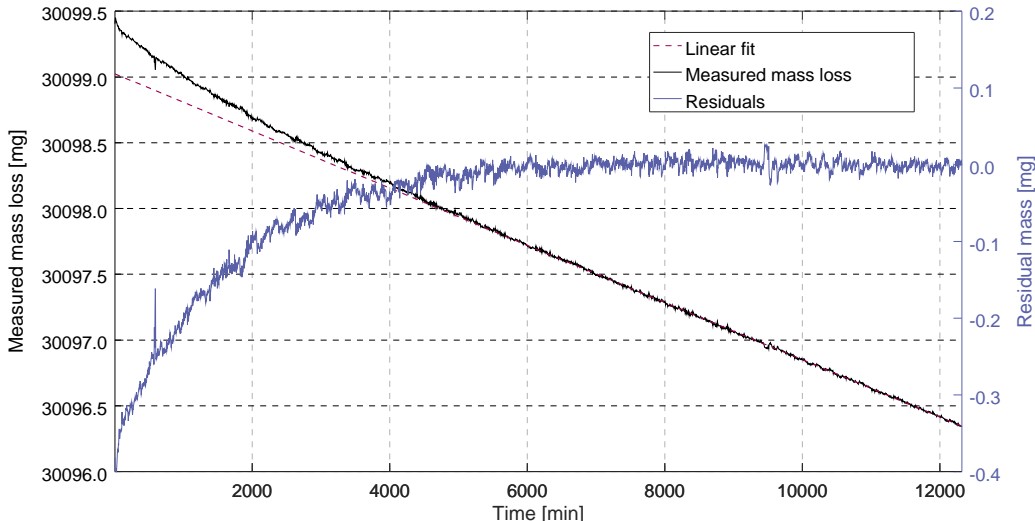

**Figure 2.** Example of measured mass loss rate for CFC-13 with magnetic suspension balance Violetta. After stabilisation of the permeation rate (first 6000 min), the mass loss becomes linear - the measured mass loss (6000 min to 12000 min) can be fitted to a linear function, yielding a mass loss rate of 217.44 ng/min. The residuals of the fit (blue line, right axis) are centered around zero and randomly distributed.

$t_{2,i} - t_{1,i}$, corresponding time difference (min);

$M_i$, molar molecular mass of substance $i$, calculated using average natural isotopic abundance (Meija et al., 2016, g/mol);

$purity_i$, purity fraction of the substance filled in permeator (mol/mol);

$V_{m,carrier}$, molar volume of carrier gas, here synthetic air, (L/mol). All volumes in this work are given at standard temperature and pressure (STP), i.e. 0°C and 1013.25 hPa. Values are from the NIST Chemistry WebBook, assuming real gas;

$q_{V1,i}$, volumetric flow of carrier gas regulated by MFC1 (Fig. 1, L/min);

$x_{res,i}$, residual molar fraction of substance $i$ in carrier gas (pmol/mol).

Permeation rates were determined at 3500 hPa and temperatures between 36°C and 45°C. We observed a particularly long stabilisation time (i.e., 6000 min) for the permeator containing CFC-13 (Fig. 2). The low mass loss for HFO-1234yf (88.5 ng/min) required a continuous measurement during 22 days to reach the required standard deviation (<0.4% for $Stab_{balance,i}$).

### 2.3 Dynamic dilution

The mixture exiting the permeation chamber, at $\mu$mol/mol level, is diluted to pmol/mol levels over two successive, dynamic dilution steps (Fig. 1). The flows are piloted by thermal MFCs (in compliance with ISO 6145-7, 2009). First, the mixture exiting the permeation chamber is diluted with a first dilution flow (Fig. 1 MFC2, up to 5 L/min). The total flow $q_{V,MFM,i}$ passing through the permeation chamber (MFC1) and diluting this flow (MFC2) is measured by a mass flow meter (MFM on Fig. 1, Vögtlin, Switzerland), so that only this MFM needs calibration to calculate the resulting molar fraction. After this first dilution step, $x_{perm,i}$ can be rewritten as:

$$x_{perm,i} = \frac{m_{1,i} - m_{2,i}}{t_{2,i} - t_{1,i}} \cdot purity_i \cdot \frac{1}{M_i} \cdot \frac{V_{m,carrier}}{q_{V,MFM,i}} + x_{res,i} \quad (2)$$

Second, a small flow of this resulting mixture is sampled by MFC3 (10 mL/min) and is diluted by another larger flow (MFC4, up to 5 L/min). The dilution factor $f_{dilution,i}$ to obtain the second dilution step is calculated as:

$$f_{dilution,i} = \frac{q_{V3,i}}{q_{V3,i} + q_{V4,i}} \quad (3)$$

with $q_{Vn,i}$, volumetric flow of MFC$_n$ (L/min) set for substance $i$.

After this second dilution step, the prepared molar fraction $x_{filled,i}$ that will be filled in all cylinders can be calculated as:

$$x_{filled,i} = x_{perm,i} \cdot f_{dilution,i} + (1 - f_{dilution}) \cdot x_{res,i} \quad (4)$$

At this stage, the generated mixture has a molar fraction approximately 10 times higher than atmospheric levels. Note that most metal surfaces in contact with the carrier gas and the produced gas mixture are passivated by applying Silco-Nert2000 coating. This includes all metal tubing, all metal surfaces of the MFCs and MFM in contact with the gas, and most of the permeation chamber.

### 2.4 Cryo-filling

The generated mixture after the second dilution step is then transferred into 11 cylinders, named MP-001 to MP-011. The

technical characteristics of these cylinders are summarised in Table 1. In brief, we used 7 cylinders made of electropolished stainless steel (Essex Industries, USA) and 4 cylinders made of Silconert2000-coated stainless steel (Swagelok). To detect potential systematic biases due to adsorption on cylinder surfaces, this set of cylinders presents four different surface/volume ratios (see details in Table 1).

A set flow of the reference gas mixture ($q_{V5,j}$, 3 L/min for the 34 L Essex cylinders, 0.5 L/min for all other smaller volume cylinders) is sampled by MFC5 (CMOSsens series, Sensirion, Switzerland, Fig. 1) and then directed to a tee. The two exiting paths of this tee are piloted by pneumatic valves (Swagelok), one being normally closed (to the cylinder, V1), the other one being normally open (to the pump, V2).

All cylinders are cleaned beforehand, being evacuated to approx. 6 Pa and filled with nitrogen at 2000 hPa (purity grade 99.999% or better), 3 times. Each cylinder was then evacuated one last time to 6 Pa or lower just before being connected to the filling system.

After being connected, the tubing between the cylinder valve and the pneumatic valve V1 (Fig. 1) is evacuated to 1 hPa and filled with synthetic air, 3 times. After leak checking, it is filled with synthetic air at a pressure of 800 hPa $\pm$ 20 hPa, because it is the residual pressure that is lost after each filling, when the cylinder valve is manually closed (see Supplement Sect. S2).

After this preparation step, the bath is (re)filled with liquid nitrogen. Once the cylinder content is re-liquified, its valve is manually opened, and, piloted by a Labview program the pneumatic valve positions are switched for a precisely set duration, in order to fill a precisely controlled volume of the reference gas mixture in each cylinder. The filling durations last from 17 min to 6 hours depending on substance and target molar fraction. To avoid freezing, the cylinder valve is intermittently heated with a heat gun during filling. Once the pneumatic valves are back at their default setting, the cylinder valve is kept open for one minute, while the tubing between the cylinder valve and V1 is heated to force all potential remaining substance of interest to be cryo-trapped in the cylinder still in liquid nitrogen. The cylinder valve is then manually closed, the cylinder disconnected and left standing vertically outside the building to warm up. A new cylinder is placed in the liquid nitrogen bath, connected to the filling system, and the filling procedure starts again. The state of the filling system is checked at least every 30 min during filling. Once the filling of each cylinder with one given mixture is completed, the permeation chamber of the balance is opened, a new permeator installed, the chamber closed, and a new mass loss measurement starts.

After the fillings of all five substances of interest are completed, additional precisely determined volumes of synthetic air are cryo-filled into the cylinders in order to dilute the mixtures to atmospheric molar fractions. For this special filling, synthetic air is also directed through the permeation chamber where a glass reaction tube filled with deionised water is

placed, evaporating water, in order to slightly humidify each cylinder. The resulting water vapour molar fraction ranges from 20 to 70 $\mu$mol/mol. Note that this added water is not included in the calculations of molar fractions for the halogenated compounds, which are therefore expressed in dry synthetic air. Adding water vapour to each cylinder was motivated by the fact that losses due to adsorption are known to occur for some halogenated compounds (Prinn et al., 2000). This has been evidenced by Yokohata et al. (1985) for $CCl_4$ and $CH_3CCl_3$ in very dry mixtures (i.e., likely less than 750 nmol/mol of water vapour), who also experimentally showed that adding water vapour to the cylinder annulled this adsorption, water vapour being an excellent competitor for adsorption sites on a metal surface (Vaittinen et al., 2013; Pogàny et al., 2016).

Reference gas mixtures for each of the five compounds were filled in this order: $SF_6$, HFC-125, HFO-1234yf, HCFC-132b and CFC-13. All cylinders are homogenised for a minimum of 6 hours each before measurement using an automated rolling system, with cylinders lying horizontally, and alternating directions. A minimum of one week elapsed between the final cylinder fillings and the measurements.

Filling different mixtures successively in each cylinder $j$ with different filling duration results in an additional dilution factor for each substance $i$. The durations are chosen in order to attain the same dilution factor for all substances in one given cylinder. The resulting molar fraction $x_{prep,i,j}$ for substance $i$ in cylinder $j$ is:

$$x_{prep,i,j} =$$
$$\frac{x_{filled,i} \cdot q_{V5,j} \cdot \Delta t_{i,j} + x_{res,i} \cdot q_{V5,j} \cdot (\Delta t_{total,j} - \Delta t_{i,j})}{\Delta t_{total,j} \cdot q_{V5,j}}$$
$$(5)$$

with the total filling duration in each cylinder $j$ being:

$$\Delta t_{total,j} = \Delta t_{SF_6,j} + \Delta t_{HFC\text{-}125,j} + \Delta t_{HFO\text{-}1234yf,j}$$
$$+ \Delta t_{HCFC\text{-}132b,j} + \Delta t_{CFC\text{-}13,j} + \Delta t_{carrier,j} \quad (6)$$

with $\Delta t_{i,j}$, filling duration of mixture containing substance $i$ in cylinder $j$;
$q_{V5,j}$, flow of MFC5 used to regulate the flow into the cylinder during cryo-filling. Equ. 5 can be simplified by removing $q_{V5,j}$, and re-arranging to:

$$x_{prep,i,j} = x_{filled,i} \cdot \frac{\Delta t_{i,j}}{\Delta t_{total,j}} + x_{res,i} \cdot \left(1 - \frac{\Delta t_{i,j}}{\Delta t_{total,j}}\right) \quad (7)$$

However, the stability component of the flow $Stab_{q_{V5,j}}$ has to be taken into account in the uncertainty budget (see Equ. 9).

Note that for a given substance, the molar fractions in cylinders vary only due to varying filling durations. This is designed to maximise the correlation between cylinders and therefore improve the resulting internal consistency (defined

in Sect. 3.2.3) of the prepared suite of reference gas mixtures. Correlation coefficients between cylinders for one given substance therefore range from 0.96 to 0.99. This set constitutes the METAS-2017 primary calibration scale for $SF_6$, HFC-125, HFO-1234yf, HCFC-132b and CFC-13. All computed prepared values are reported in Table 3.

## 2.5 Uncertainty of preparation

### 2.5.1 Uncertainty of prepared values in pmol/mol

We estimate the uncertainty of the assigned molar fraction, for each substance in each cylinder, following JCGM:2008 by measuring (type A uncertainty) or assigning (type B uncertainty) an uncertainty estimate to each input quantity used in the equations presented in the Method. Expanded uncertainties, noted U, are then calculated with $k = 2$ for a coverage probability of approximately 95 %. All uncertainty computations for the prepared molar fractions are made using the GUMWorkBench program. We describe hereafter the most important contributions to the uncertainty.

**Balance measurement**: The uncertainty of the weighing is estimated by fitting a linear function through the measured masses over time, and using the standard deviation between this linear fit and the residuals for each point as uncertainty estimate ($k = 1$, e.g. Fig. 2). This measurement noise level, noted $Stab_{balance,i}$ in Equ. 8, is in the order of 0.2 % of the mass loss (for $SF_6$, whose mass loss was higher making the associated balance noise relatively smaller) to 0.6 %. In addition, we take into account potential biases due to mass calibration of the two used calibration masses (CM1, CM2) and time uncertainty, even if these two contributions are extremely small (see also Sect. S1 in the Supplement).

**Permeation chamber temperature stability** $Stab_T$: Once carrier gas flow and pressure are kept constant, the permeation rate varies only with temperature. The stability of the permeation chamber temperature is 0.02°C over 20 min ($k = 2$). Based on our experience measuring temperature sensitivity of permeation rate, this corresponds to approx. 0.1 % change in permeation rate. $Stab_{balance,i}$ and $Stab_T$ are given a value of one and included as follow to Equ. 2 in order to take into account their uncertainty contributions:

$$x_{perm,i} = \frac{m_{1,i} - m_{2,i}}{t_{2,i} - t_{1,i}} \cdot Stab_{balance,i} \cdot Stab_T \cdot purity_i$$
$$\cdot \frac{1}{M_i} \cdot \frac{V_{m,carrier}}{q_{V,MFM,i}} + x_{res,i} \quad (8)$$

**Substance purity** $purity_i$: We use the certificate provided by the substance manufacturer, i.e. $purity_i = 0.999$ for $SF_6$ and $purity_i = 0.99$ for all other substances. As uncertainty we choose a triangular distribution with 1 as upper boundary, and $1-2\cdot(1-purity_i)$ as lower boundary. This is a conservative approach.

**Impurity in the carrier gas** $x_{res,i}$: For all fillings, a total of three 50 L cylinders of the same type of synthetic air (Pan-

gas synthetic air 5.6) have been used. Absolute impurity levels in one cylinder were measured at Empa on the Medusa-GC-MS system (Miller et al., 2008). The similarity of the impurity level in the other cylinders was checked at METAS on a similar preconcentration-GC-MS system, by trapping a total of 6 litres of synthetic air. For each measured impurity molar fraction $x_{res,i}$ (ranging from 6 to 30 fmol/mol), we use a triangular distribution centred in $x_{res,i}$, with zero as lower boundary and $x_{res,i}\cdot2$ as upper boundary (see Table S4).

**Calibrated values of volumetric flows**: $q_{V,MFM,i}$, $q_{V3,i}$ and $q_{V4,i}$ (MFM, MFC3 and MFC4 in Fig. 1) were calibrated using a SI-traceable primary reference standard applying a system of pistons with known volume (Niederhauser and Barbe, 2002). All flows are given at standard temperature and pressure (0°C, 1013.25 hPa). The actual flow has been calibrated using the same gas type, at the same input parameter values (pressure set points before and after each MFM/MFC, MFC set points). A minimum of 4 replicate measurements for each flow set point have been made and the average was taken as best estimate. The stability of each flow over the measurement duration (1-5 minutes) is better than 1 permil. The obtained, relative expanded uncertainty ($k = 2$) for each flow is U = 0.3 % for MFC3 (due to smaller flows) and U = 0.2 % for MFM and MFC4.

**Filling durations** $\Delta t_i$: Individual filling durations range from 1060 s to 24110 s (18 min to 6.7 hours, see Supplement Table S3). The associated uncertainty is fixed at U = 1.8 s and takes into account the response time of each pneumatic valve as well as the computer time clock uncertainty. In percentage this uncertainty therefore decreases with increasing filling duration.

**Stability of filling flow** $Stab_{q_{V5,j}}$: The flow stability of MFC5 was challenged due to the relatively small pressure gradient of approx. 800 hPa. We therefore take into account a flow stability component in the uncertainty of U = 0.1 %. This stability component plays a role to explain the internal consistency between cylinders (Sect. 3.2.3).

The resulting, combined uncertainty is then expanded using a coverage factor $k = 2$ (representing a 95 % confidence interval for a Gaussian distribution). The obtained values are documented in Table 3. As an example for cylinder MP-001, we present in Fig. 3 the uncertainty contribution for the most important contributors as pie charts, for each substance. Expanded uncertainties range from 0.6 % (for $SF_6$) to 1.3 %. The smallest uncertainty obtained for $SF_6$ is mostly explained by the high substance purity level inside the permeator (0.999 pure, ten times better than for the other substances), as well as low $SF_6$ contamination of the carrier gas.

### 2.5.2 Uncertainty of prepared ratios

To calculate the expected internal consistency of the prepared suite of mixtures, we also calculate ratios of assigned values, using cylinder MP-001 as master cylinder. The assigned value of a ratio for substance $i$ between cylinder $j$ and cylin-

**Table 2.** Primary reference gas mixtures for $SF_6$ filled in cylinder MP-001: List of variables taken into account in the uncertainty budget. Variables and corresponding numbers in italic are intermediate results. To fill $SF_6$ in the other cylinders, only the filling durations were modified (i.e. last section in this Table), all other input values remained unchanged. Input values used to calculate molar fractions and expanded uncertainties for the other substances can be found in the Supplement, Tables S1, S2 and S3.

| Quantity | Value | Standard uncertainty ($k = 1$) | Unit | Distribution | Sensitivity coefficient | Uncertainty contribution in: pmol/mol | % |
|---|---|---|---|---|---|---|---|
| **Weighing of $SF_6$ permeation device** | | | | | | | |
| $m_{1,SF_6}$ | 28.44957646 | 0.000000289 | g | rectangular | 790 | 2.30E-004 | 0 |
| $m_{2,SF_6}$ | 28.43612094 | 0.000000289 | g | rectangular | -790 | -2.30E-004 | 0 |
| $t_{1,SF_6}$ | 5999.45 | 0.025 | min | normal | 1.30E-003 | 3.30E-005 | 0 |
| $t_{2,SF_6}$ | 14130.8 | 0.025 | min | normal | -1.30E-003 | -3.30E-005 | 0 |
| $Stab_{balance,SF_6}$ | 1 | 0.002 | - | normal | 11 | 0.021 | 46.3 |
| $Permeation_{SF_6}$ | *1654.77* | *3.31* | *ng/min* | | | | |
| $Stab_T$ | 1 | 0.0005 | - | normal | 11 | 5.30E-003 | 2.9 |
| $Purity_{SF_6}$ | 0.9995 | 0.000204 | - | triangular | 11 | 2.20E-003 | 0.5 |
| **Molar volume of carrier gas** | | | | | | | |
| $fraction_{O2}$ | 0.2 | 0.005 | - | normal | -6.10E-003 | -3.10E-005 | 0 |
| $d_{O2}$ | 1.4287 | 0.00015 | kg/m$^3$ | normal | -1.6 | -2.50E-004 | 0 |
| $d_{N2}$ | 1.2501 | 0.00015 | kg/m$^3$ | normal | -6.6 | -9.90E-004 | 0.1 |
| $M_O$ | 15.9994 | 0.000151 | g/mol | triangular | 0.15 | 2.20E-005 | 0 |
| $M_N$ | 14.006855 | 0.000174 | g/mol | triangular | 0.59 | 1.00E-004 | 0 |
| $V_{m,carrier}$ | *28.8107* | *0.0199* | *L/mol* | | | | |
| **Molar mass of $SF_6$** | | | | | | | |
| $M_S$ | 32.0675 | 0.00347 | g/mol | triangular | -0.072 | -2.50E-004 | 0 |
| $M_F$ | 18.998403163 | 0.000000003 | g/mol | normal | -0.43 | -1.30E-009 | 0 |
| $M_{SF_6}$ | *146.05792* | *0.00347* | *g/mol* | | | | |
| **Dynamic dilution** | | | | | | | |
| $q_{V,carrier,SF_6}$ | 4594.97 | 4.59 | mL/min | normal | -2.30E-003 | -0.011 | 11.6 |
| $q_{V3,SF_6}$ | 10.809 | 0.0162 | mL/min | normal | 0.98 | 0.016 | 25.9 |
| $q_{V4,SF_6}$ | 5115.46 | 5.12 | mL/min | normal | -2.10E-003 | -0.011 | 11.5 |
| $f_{SF_6}$ | *2.11E-003* | *3.79E-006* | - | | | | |
| $x_{res,SF_6}$ | 6.00E-003 | 2.45E-003 | pmol/mol | triangular | 1 | 2.40E-003 | 0.6 |
| $x_{filled,SF_6}$ | *116.438* | *0.34* | *pmol/mol* | | | | |
| **Cryo-filling in cylinder MP-001** | | | | | | | |
| $\Delta t_{SF_6,1}$ | 4018 | 0.9 | s | normal | 2.40E-003 | 2.20E-003 | 0.5 |
| $\Delta t_{HFC125,1}$ | 4018 | 0.9 | s | normal | -2.40E-004 | -2.20E-004 | 0 |
| $\Delta t_{HFC1234yf,1}$ | 4018 | 0.9 | s | normal | -2.40E-004 | -2.20E-004 | 0 |
| $\Delta t_{HCFC132b,1}$ | 4018 | 0.9 | s | normal | -2.40E-004 | -2.20E-004 | 0 |
| $\Delta t_{CFC13,1}$ | 4018 | 0.9 | s | normal | -2.40E-004 | -2.20E-004 | 0 |
| $\Delta t_{water,1}$ | 24110 | 0.9 | s | normal | -2.40E-004 | -2.20E-004 | 0 |
| $\Delta t_{total,1}$ | *44200* | *4.58* | *s* | | | | |
| $Stab_{MFC5}$ | 1 | 0.0005 | - | normal | -0.96 | -9.60E-004 | 0 |
| $x_{prep,SF_6,1}$ | *10.590* | *0.032* | *pmol/mol* | | | | |

der MP-001 (marked by subscript 1 hereafter) can be calculated with a very good approximation by:

$$R_{prep,i,j} = \frac{Stab_{T,i,j}}{Stab_{T,i,1}} \cdot \frac{Stab_{qV5,j}}{Stab_{qV5,1}} \cdot \frac{\Delta t_{total,1}}{\Delta t_{total,j}} \cdot \frac{\Delta t_{i,j}}{\Delta t_{i,1}} \quad (9)$$

The terms 'Stab' represent the stability of each filling temperature and each filling flow of MFC5, respectively. Each

'Stab' term is assigned a value of 1 and an expanded uncertainty of $Stab_{T,i,j} = 0.1$ % for the temperature stability and $Stab_{qV5,j} = 0.1$ % for the flow stability (as discussed in Sect. 2.5.1). The standard uncertainty ($k = 1$) of $R_{prep,i,j}$, calculated using Equ. 9, is hereafter noted $u_{R_{prep,i,j}}$.

Ratio values range from 0.8 to 1.3 and the corresponding expanded uncertainty has an actually constant value of 0.3 %

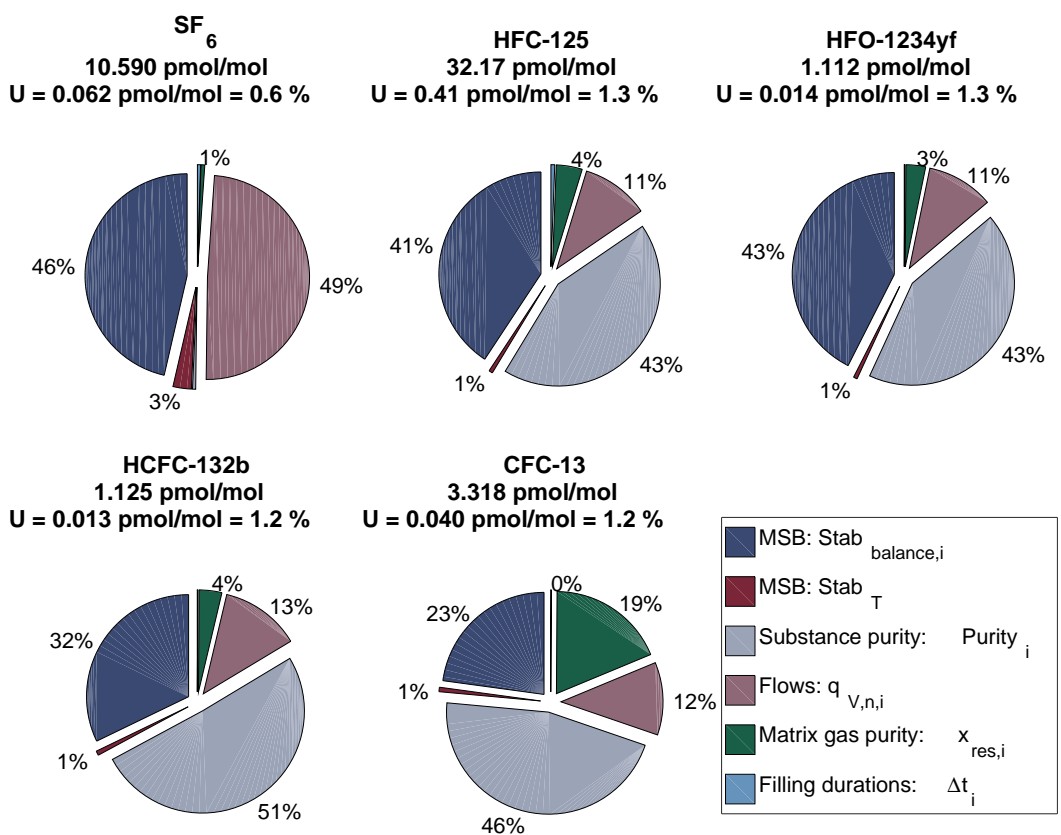

**Figure 3.** Uncertainty budget of the preparation for cylinder MP-001. The pie charts depict the most important sources of uncertainties, in percent, for each substance (see also Sect. 2.5.1). U is the relative expanded uncertainty of the preparation ($k = 2$). All input values used to calculate the budget for $SF_6$ in MP-001 are detailed in Table 2.

over this limited ratio range (Table 3). Due to the elimination of many common factors when working in such a ratio space, the correlation coefficient between ratios is approximately 0.4.

## 3 Measurement results and discussion

### 3.1 Measurement method

The relative molar fractions of the five compounds in the eleven samples MP-001 – MP-011 were determined using Medusa gas chromatography mass-spectrometry (GC-MS) methods (Miller et al., 2008; Arnold et al., 2012). Medusa GC-MS systems have been in use in AGAGE for hourly field and laboratory measurements of more than 50 halogenated compounds. In brief, it consists of a multi-port inlet, a sample drying system (nafion driers), a custom-designed preconcentration system, a capillary GC column (CP-PoraBOND Q, 0.32 mm ID x 25 m, 5 $\mu$m film thickness, Varian Chrompack) and a quadrupole mass spectrometer (Agilent Technologies, 5975).

Measurements were conducted on the Empa laboratory Medusa GC-MS and consisted of 2 L samples measured alternatingly with a reference gas measurement. MP-001 was used as the reference sample. Hence all results of the MP-002 – MP-011 samples are expressed as ratios by dividing the chromatographic peak area of the sample by the mean of the area of the bracketing MP-001 measurements. Minor analytical modifications compared to the routine field measurements were adopted. In particular, to enhance precision of the measurements, compounds, which chromatographically elute near those of interest, were omitted from acquisitions, and the electron multiplier was elevated to enhance the signal size. Integration of the chromatographic peaks was done using commercial software (GCWerks). Detection limits, defined here as three times the noise level, are 0.015 pmol/mol for $SF_6$, 0.02 pmol/mol for HFC-125, 0.01 pmol/mol for HFO-1234yf, 0.015 pmol/mol for HCFC-132b and 0.07 pmol/mol for CFC-13. All results are expressed as dry-air molar fraction. The system was shown to be free of blanks and linear in the range of molar fractions of our samples (see e.g. Vollmer et al., 2015). The repeatability of the measurements was calculated as the standard deviation (1 $\sigma$) of the area-ratios. The standard deviation of the mean was obtained by dividing by the square root of n, n being the number of measurements.

**Table 3.** METAS-2017 suite of reference gas mixtures: prepared ratios and molar fractions and associated uncertainty with a 95% confidence interval. Molar fractions for halogenated compounds are expressed in dry synthetic air. Ratios represent the number of moles of substance $i$ in cylinder $j$ divided by the number of moles of the same substance in cylinder MP-001. $R_{prep,i,j}$ is therefore a molar ratio. Prepared ratios are identical for all substances within a given cylinder and thus only one value per cylinder is reported here. Note: HFO-1234yf was not filled in cylinder MP-011.

| Cylinder | 6 | 10 | 8 | 1 & 7 | 11 | 3 & 4 | 2 & 9 | 5 |
|---|---|---|---|---|---|---|---|---|
| | | | **SF$_6$** | | | | | |
| Molar fraction, pmol/mol | 8.47 | 9.53 | 10.48 | 10.59 | 10.70 | 11.12 | 11.65 | 13.77 |
| U, pmol/mol | 0.05 | 0.06 | 0.07 | 0.06 | 0.07 | 0.07 | 0.07 | 0.08 |
| U, % | 0.6 | 0.6 | 0.6 | 0.6 | 0.6 | 0.6 | 0.6 | 0.6 |
| | | | **HFC-125** | | | | | |
| Molar fraction, pmol/mol | 25.74 | 28.95 | 31.84 | 32.17 | 32.49 | 33.77 | 35.38 | 41.82 |
| U, pmol/mol | 0.33 | 0.37 | 0.40 | 0.41 | 0.41 | 0.43 | 0.44 | 0.52 |
| U, % | 1.3 | 1.3 | 1.3 | 1.3 | 1.3 | 1.3 | 1.2 | 1.2 |
| | | | **HFO-1234yf** | | | | | |
| Molar fraction, pmol/mol | 0.890 | 1.001 | 1.101 | 1.112 | - | 1.167 | 1.223 | 1.445 |
| U, pmol/mol | 0.011 | 0.013 | 0.014 | 0.014 | - | 0.015 | 0.015 | 0.018 |
| U, % | 1.2 | 1.3 | 1.3 | 1.3 | - | 1.3 | 1.2 | 1.2 |
| | | | **HCFC-132b** | | | | | |
| Molar fraction, pmol/mol | 0.900 | 1.012 | 1.113 | 1.125 | 1.136 | 1.181 | 1.237 | 1.462 |
| U, pmol/mol | 0.011 | 0.012 | 0.013 | 0.013 | 0.013 | 0.014 | 0.014 | 0.017 |
| U, % | 1.2 | 1.2 | 1.2 | 1.2 | 1.1 | 1.2 | 1.1 | 1.2 |
| | | | **CFC-13** | | | | | |
| Molar fraction, pmol/mol | 2.657 | 2.987 | 3.284 | 3.318 | 3.351 | 3.482 | 3.648 | 4.310 |
| U, pmol/mol | 0.034 | 0.037 | 0.040 | 0.040 | 0.041 | 0.042 | 0.044 | 0.051 |
| U, % | 1.3 | 1.2 | 1.2 | 1.2 | 1.2 | 1.2 | 1.2 | 1.2 |
| | | | **Ratio** | | | | | |
| **Prepared ratio, mol/mol** | **0.800** | **0.900** | **0.990** | **1.000** | **1.010** | **1.050** | **1.100** | **1.300** |
| U, mol/mol | 0.002 | 0.003 | 0.003 | 0.003 | 0.003 | 0.003 | 0.003 | 0.004 |
| U, % | 0.3 | 0.3 | 0.3 | 0.3 | 0.3 | 0.3 | 0.3 | 0.3 |

**Table 4.** Correcting the measured ratio: results of weighted linear fitting and calculated internal consistency within the METAS-2017 suite of reference gas mixtures, for each substance. Values correspond to calculations done after exclusion of the outliers.

| | SF$_6$ | HFC-125 | HFO-1234yf | HCFC-132b | CFC-13 |
|---|---|---|---|---|---|
| | | Fitting $R_{meas,i,j} = a_i \cdot R_{prep,i,j} + b_i$ | | | |
| $a_i$ | 1.0217 | 1.0123 | 1.0182 | 1.0013 | 1.0204 |
| $u_{a_i}$ | 0.0040 | 0.0042 | 0.0074 | 0.0058 | 0.0075 |
| $b_i$ | -0.0210 | -0.0078 | -0.0229 | -0.0026 | -0.0293 |
| $u_{b_i}$ | 0.0015 | 0.0017 | 0.0032 | 0.0025 | 0.0029 |
| Internal consistency, 1 $\sigma$, % | 0.23 | 0.35 | 1.1 | 0.24 | 0.6 |
| | | Fitting $R_{meas,i,j} = a_i \cdot R_{prep,i,j}$ | | | |
| $a_i$ | 1.0019 | 1.0050 | 0.9974 | 0.9989 | 0.9931 |
| $u_{a_i}$ | 0.0039 | 0.0042 | 0.0073 | 0.0058 | 0.0073 |

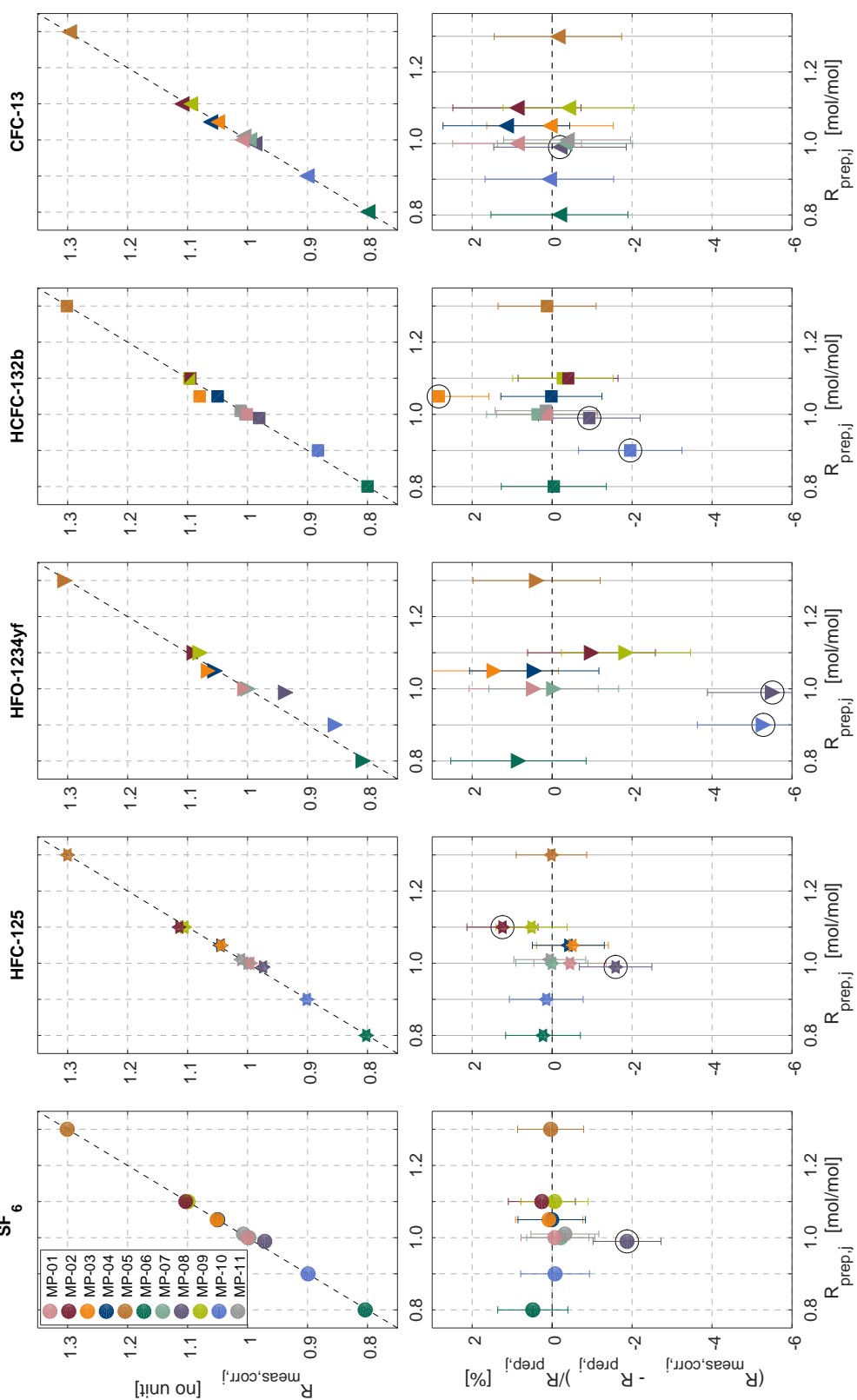

**Figure 4.** Internal consistency estimates for the METAS-2017 primary calibration scales. Top: Measured ratios $R_{meas,corr,i,j}$ vs prepared ratios $R_{prep,i,j}$. $R_{prep,i,j}$ represents the prepared molar fraction in cylinder $j$ divided by the corresponding prepared molar fraction in cylinder MP-001 (Equ. 9). The unit of $R_{prep,i,j}$ is therefore mol/mol. $R_{meas,corr,i,j}$ is a signal ratio and unit-less. Bottom: difference, in percent, between prepared and measured ratios. The error bars take into account the uncertainty of $R_{prep,i,j}$ as well as the measurement standard deviation of the mean and represent a 95% confidence interval. Outliers excluded from the calculation of the analyser response function are highlighted by black open circles.

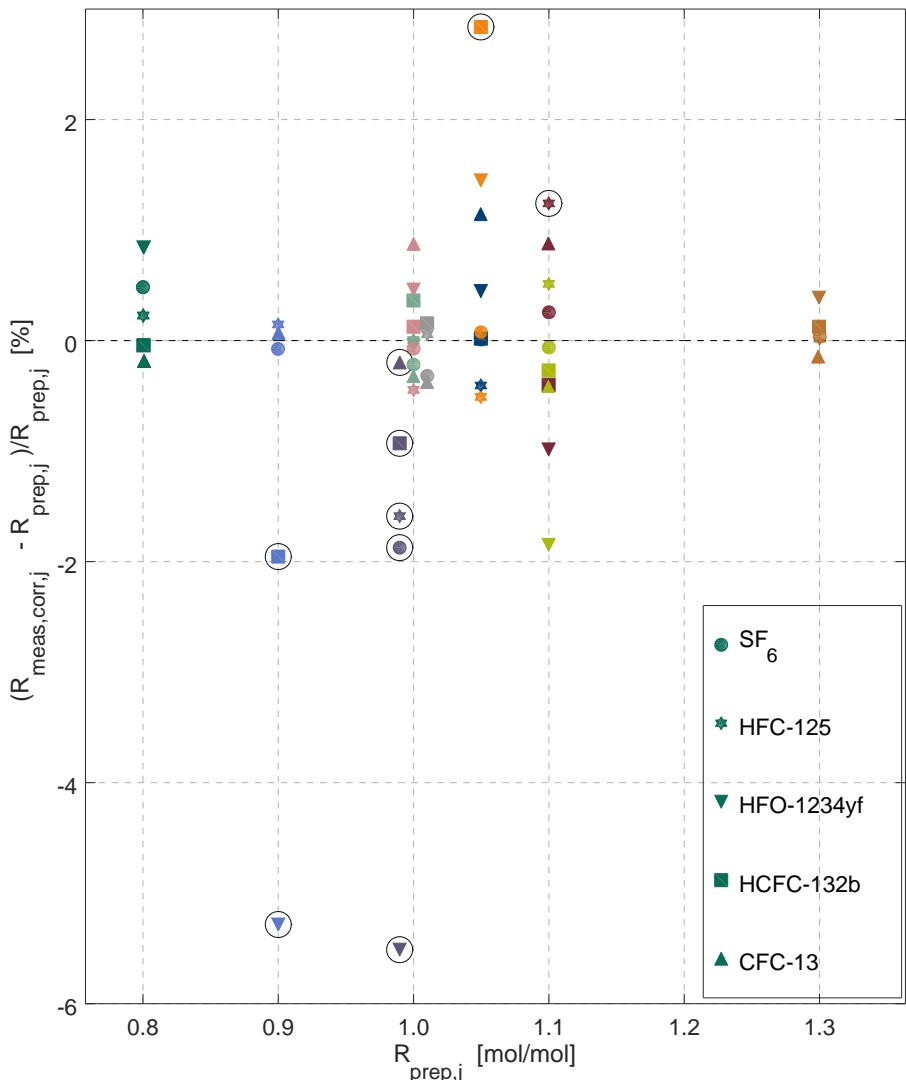

**Figure 5.** Overview of internal consistency estimates for the METAS-2017 calibration scales. Prepared and measured ratios of molar fractions for each substance (shown by different marker types) in each cylinder (shown using different colors, see color legend in Fig. 4). Outliers are highlighted by black open circles (see main text Sect. 3.2.2).

## 3.2   Measurement results

Quantifying analyser response, correcting results and identifying potential outliers is done in an iterative manner. First, all measurement results with their standard deviation are used to calculate a first analyser response function. Using this function, all measurements results are corrected. The corrected values are then compared to the assigned value. Based on the results of this test, potential outliers are excluded, and an adjusted analyser response is calculated, etc. Details of the calculations are presented hereafter.

## 3.2.1   Analyser response calibration

Following the approach already developed at SIO, and similar to methods already used for isotopic studies (e.g., Dansgaard, 1953; Craig, 1957), we compare measured and assigned values in the ratio space, for three reasons. First, the mass spectrometer used for analysis is naturally drifting over time and to correct for this effect, measured areas are expressed as area ratios relative to the bracketing mixture used as standard, here MP-001. The most precise measurement result given by the MS is therefore a ratio of areas. Sec-

ond, due to the preparation design made to maximise correlation between cylinders for one given substance, here again the prepared ratio $R_{prep,i,j}$ has an uncertainty much smaller than each molar fraction separately, because when calculating these ratios many constant factors cancel out (Equ. 9). The uncertainty components that still have to be taken into account are mostly those related to the stability over time of the preparation system, i.e. the stability of the permeation temperature, the stability of each MFC flow (negligible, except for MFC5), and the filling duration uncertainty. Expanded uncertainties in ratio space are therefore 0.3 %, compared to expanded uncertainties of 0.6 % to 1.3 % in the molar fraction space (Table 3). Third, the correlation between values of assigned ratios is much smaller (in the order of 0.4) than the correlation between assigned molar fraction values, and therefore using ratios is again better indicated to estimate the fit between assigned values and instrument response (Sect. 5.4.2 in ISO 6143, 2001).

To calibrate the measured value, we thus determine the analyser response function in ratio space, i.e. measured ratio vs assigned ratio. We can therefore compare a maximum of 10 ratio values, using 11 cylinders. An example is given in Fig. 4.

The analyser response is calculated using a linear fit due to the relatively small number of measured values as well as the good linear response of the MS over this limited range:

$$R_{meas,i,j} = a_i \cdot R_{prep,i,j} + b_i \qquad (10)$$

with $R_{meas,i,j}$, the measured area ratio for substance $i$ in cylinder $j$.

The fit coefficients $a_i$ and $b_i$ are computed using a bivariate weighted linear fit, following the York algorithm (York et al., 2004) as described in Cantrell (2008) (Supplement, Sect. S3), coded in Octave (results in Table 4). As an additional test we ran the same fitting algorithm forcing $b_i$=0. Interestingly, for each substance the obtained slope can then not be distinguished from a slope of 1 within uncertainties (Table 4). This suggests that the analyser response is linear within the tested range and within stated uncertainties. The deviation in the corrected values for cylinder MP-001 is no more than 0.2 %, but varies up to 0.5 % for MP-005 and 1 % for MP-006, which are at the upper and lower end of the scale. We also calculated $a_i$ and $b_i$ using a simpler weighted linear fit considering only weights associated with the measured values, giving very similar results (difference < 0.03 %).

The measured & corrected ratios $R_{meas,corr,i,j}$ and values in pmol/mol $x_{meas,corr,i,j}$ are calculated using:

$$R_{meas,corr,i,j} = \frac{R_{meas,i,j} - b_i}{a_i} \qquad (11)$$

$$x_{meas,corr,i,j} = R_{meas,corr,i,j} \cdot x_{prep,i,1} \qquad (12)$$

with $x_{prep,i,1}$, the prepared value in pmol/mol for substance $i$ in cylinder MP-001.

Hereafter we refer to $R_{meas,corr,i,j}$ as measured ratios and to $x_{meas,corr,i,j}$ as measured molar fractions.

### 3.2.2 Verification test and exclusion of outliers

To compare the measured results to the prepared results, we use the measured ratios $R_{meas,corr,i,j}$ and their associated uncertainties $u_{R_{meas,corr,i,j}}$ calculated according to Equ. 11 (Supplement, Table S5). We then calculate the verification criteria (Sect. 5.2.2 in ISO 6143, 2001):

$$d_{i,j} = |R_{meas,corr,i,j} - R_{prep,i,j}| \\ - \left( 2 \cdot \sqrt{u^2_{R_{meas,corr,i,j}} + u^2_{R_{prep,i,j}}} \right) \leq 0 \quad (13)$$

Using this procedure, after the first iteration cylinder MP-008 is excluded as outlier for $SF_6$, HFC-125 and HFO-1234yf, the measured value being systematically too low. Because this represents already the majority of all substances, we decided to exclude this cylinder for HCFC-132b and CFC-13 as well. After the second iteration cylinder MP-002 is excluded for HFC-125, MP-010 for HFO-1234yf and MP-003 for HCFC-132b. After the third iteration, cylinder MP-010 is identified as an outlier for HCFC-132b.

For cylinder MP-008 (a 4.5 L Essex cylinder filled at 24 bar, Table 1), we note that the difference between measured value and assigned value is large for $SF_6$ (the first filled substance) and more or less decreases until showing no particular offset for CFC-13 (being filled last). An explanation for such a time varying offset could be a potential leak over time, affecting more the substance that was filled first.

During filling of cylinder MP-002 for HFC-125, we noted that the mass flow controller sampling the flow into the cylinder (MFC5, Fig. 1) was suffering from large instability during the 15 s towards the end of the filling. This particularly high instability was very likely due to both the small pressure gradient over MFC5, and additionally the small flow from MFC4 (4 L/min) compared to the flow sampled by MFC5 (3 L/min), making the pressure control before MFC5 (done by PrC2) to be particularly challenging. To improve the pressure regulation by PrR2 and therefore limit MFC5 instability, an additional buffer volume (approx. 1 L, stainless steel) was added on the flow path just before PrC2. This instability did not occur during subsequent fillings.

We tentatively explain cylinder MP-003 being an outlier for HCFC-132b by the fact that this cylinder was filled first after a synthetic air cylinder exchange. Additional tests showed that HCFC-132b is affected by regulator contamination and takes time to clear out. The regulator was purged three times before using the synthetic air but perhaps this was not sufficient. The system being then continuously running, the purge was already sufficient for the second filling not to be affected.

We unfortunately do not have a specific explanation for cylinder MP-010 being an outlier for HCFC-132b and HFO-1234yf, potentially pointing towards operator error,

and highlighting the need of a verification step. We however observe that most outliers are cases of substance loss (Fig. 5) and affect cylinders having the smallest total amount of gas filled and the highest surface/volume ratio (i.e. Essex cylinder 4.5 L, 24 bar). We would therefore in the future favour filling in cylinders of larger volume and pressure, as well as a further automatising the cryo-filling process to limit human intervention as much as possible as well as to increase the safety of the procedure. Furthermore, we plan to investigate if the observed substance losses occurred by adsorption on cylinders walls or beforehand in the preparation system. To do so, a comparison of the molar fraction in the mixture exiting the magnetic suspension balance/dynamic dilution system with the same mixture filled in cylinders will be performed. The recent installation of a measurement system for halogenated gases at METAS in the same laboratory makes this possible. If the adsorption indeed occurs in the cylinders, it will be tested if adding water vapour earlier in the sequence of fillings may help to limit this adsorption.

After excluding the outliers, we can observe no systematic bias between mixtures filled in electro-polished cylinders from those filled in SilcoNert-coated cylinders. This result gives us confidence that there is also no identifiable loss of the halogenated substances on cylinder surfaces.

### 3.2.3 Internal consistency of the suite

To give an estimation of the preparation reproducibility, we calculate the so-called 'internal consistency' of the suite of mixture (e.g., Prinn et al., 2000). This parameter quantifies the difference between assigned values and measured values, for each substance.

First, we calculate the difference $d$ for each substance, in each cylinder:

$$d_{i,j}[\%] = \frac{x_{meas,corr,i,j} - x_{prep,i,j}}{0.5 \cdot (x_{meas,corr,i,j} + x_{prep,i,j})} \cdot 100 \qquad (14)$$

and the associated uncertainty:

$$u_{d_{i,j}}[\%] = \frac{\sqrt{u_{x_{meas,corr,i,j}}^2 + u_{x_{prep,i,j}}^2}}{0.5 \cdot (x_{meas,corr,i,j} + x_{prep,i,j})} \cdot 100 \qquad (15)$$

We then calculate the weighted mean difference for each substance, over the set of cylinders (without outliers):

$$d_{W,i}[\%] = \frac{\sum W_{d_{i,j}} \cdot d_{i,j}}{\sum W_{d_{i,j}}} \quad with \quad W_{d_{i,j}} = \frac{1}{u_{d_{i,j}}^2} \qquad (16)$$

We use the corresponding weighted standard deviation $u_{d_{W,i}}$ of $d_{W,i}$ as estimator of the internal consistency:

$$u_{d_{W,i}}[\%] = \sqrt{\frac{\sum_{j=1}^{N} W_{d_{i,j}}(d_{i,j} - d_{W,i})^2}{\frac{N-1}{N} \cdot \sum_{j=1}^{N} W_{d_{i,j}}}} \qquad (17)$$

The calculated internal consistencies for each substance are reported in Table 4. This measured internal consistency

is due to measurement reproducibility of the Medusa-GC-MS system as well as how stable the preparation system is (i.e., including all potential sources of random noise in the preparation system, any systematic bias being on the contrary canceled out when working in ratio space).

### 3.3 Final assigned uncertainties

#### 3.3.1 According to ISO-6142-1:2015

According to ISO 6142-1 (2015), the expanded uncertainty ($k = 2$) of the final molar fraction after the verification step can be calculated as:

$$U_{x_{meas,corr,i,j},ISO} = \sqrt{u_{prep,i,j}^2 + u_{meas,corr,i,j}^2 + (x_{prep,i,j} - x_{meas,corr,i,j})^2} \qquad (18)$$

This formula includes the uncertainty of the preparation as well as the uncertainty of the verification, with equal weights. The resulting uncertainties range from 0.7 % for $SF_6$ to 1.5 % for HFO-1234yf (excluding outliers). In particular, here the uncertainties for HFC-125 and HCFC-132b become smaller (1 %) than the prepared uncertainty (1.3 %), because in this formula uncertainties from preparation and measurement are arbitrarily given equal weights.

#### 3.3.2 According to preparation and measurement equations

We calculate the uncertainty of $x_{meas,corr,i,j}$ using the uncertainty of each component defining it, according to Equ. 12. This method therefore includes both the preparation uncertainty (through $x_{prep,i,1}$) and the measurement uncertainty (through $R_{meas,corr,i,j}$). The expanded uncertainty ($k = 2$) ranges from 1 % for $SF_6$ to 2 % for HCFC-132b and CFC-13 (Table 5). The disadvantage of using this method is that $u_{R_{prep,i,j}}$ is included twice, once in $u_{R_{meas,corr,i,j}}$ through $a_i$ and $b_i$, and once in $u_{x_{prep,i,1}}$ that includes the same stability factors as in $u_{R_{prep,i,j}}$ (Equ. 2 and 7). To correct for this effect we would need to remove the stability factors in $u_{x_{prep,i,1}}$. However, the uncertainty budgets of the prepared mixtures (Fig. 3) suggest that this contribution is overall minor (from less than 1 % to 5 % of the total), and removing one occurrence of $u_{R_{prep,i,1}}$ would be a marginal change.

We favour this method to calculate the uncertainty, where the sensitivity of the final value to the preparation and verification uncertainties are taken into account, through Equ. 12, and where the resulting uncertainty is the largest – it is also the most conservative approach.

**Table 5.** METAS-2017 suite of reference gas mixtures: final molar fraction values $x_{meas,corr,i,j}$ expressed in dry synthetic air and associated uncertainties at a 95% confidence interval.

| Cylinder | 6 | 10 | 8 | 1 | 7 | 11 | 3 | 4 | 2 | 9 | 5 |
|---|---|---|---|---|---|---|---|---|---|---|---|
| **SF$_6$** | | | | | | | | | | | |
| Molar fraction, pmol/mol | 8.513 | 9.522 | 10.286 | 10.582 | 10.566 | 10.663 | 11.126 | 11.119 | 11.679 | 11.642 | 13.774 |
| U, pmol/mol | 0.084 | 0.094 | 0.101 | 0.104 | 0.105 | 0.105 | 0.110 | 0.110 | 0.115 | 0.115 | 0.136 |
| U, % | 1.0 | 1.0 | 1.0 | 1.0 | 1.0 | 1.0 | 1.0 | 1.0 | 1.0 | 1.0 | 1.0 |
| **HFC-125** | | | | | | | | | | | |
| Molar fraction, pmol/mol | 25.801 | 28.992 | 31.337 | 32.027 | 32.168 | 32.514 | 33.597 | 33.632 | 35.826 | 35.566 | 41.828 |
| U, pmol/mol | 0.392 | 0.442 | 0.477 | 0.488 | 0.490 | 0.495 | 0.512 | 0.513 | 0.546 | 0.541 | 0.639 |
| U, % | 1.5 | 1.5 | 1.5 | 1.5 | 1.5 | 1.5 | 1.5 | 1.5 | 1.5 | 1.5 | 1.5 |
| **HFO-1234yf** | | | | | | | | | | | |
| Molar fraction, pmol/mol | 0.898 | 0.949 | 1.042 | 1.117 | 1.111 | - | 1.184 | 1.173 | 1.211 | 1.201 | 1.451 |
| U, pmol/mol | 0.018 | 0.018 | 0.020 | 0.022 | 0.022 | - | 0.023 | 0.023 | 0.024 | 0.024 | 0.029 |
| U, % | 2.0 | 1.9 | 1.9 | 2.0 | 2.0 | - | 2.0 | 2.0 | 2.0 | 2.0 | 2.0 |
| **HCFC-132b** | | | | | | | | | | | |
| Molar fraction, pmol/mol | 0.900 | 0.993 | 1.103 | 1.126 | 1.129 | 1.138 | 1.215 | 1.181 | 1.232 | 1.234 | 1.464 |
| U, pmol/mol | 0.015 | 0.016 | 0.018 | 0.018 | 0.019 | 0.019 | 0.020 | 0.019 | 0.020 | 0.020 | 0.024 |
| U, % | 1.6 | 1.6 | 1.6 | 1.6 | 1.6 | 1.6 | 1.6 | 1.6 | 1.6 | 1.6 | 1.6 |
| **CFC-13** | | | | | | | | | | | |
| Molar fraction, pmol/mol | 2.652 | 2.989 | 3.278 | 3.347 | 3.307 | 3.339 | 3.484 | 3.523 | 3.681 | 3.634 | 4.304 |
| U, pmol/mol | 0.052 | 0.057 | 0.065 | 0.065 | 0.067 | 0.064 | 0.067 | 0.068 | 0.072 | 0.072 | 0.085 |
| U, % | 2.0 | 1.9 | 2.0 | 1.9 | 2.0 | 1.9 | 1.9 | 1.9 | 2.0 | 2.0 | 2.0 |

## 4 Comparison to other existing reference gas mixtures

Cylinder MP-001 was used to compare the METAS-2017 calibration scale to other scales, using Empas Medusa-GC-MS as comparator. We report hereafter a brief description of each calibration scale followed by the results of the comparisons, for SF$_6$, HFC-125, HFO-1234yf and CFC-13. For HCFC-132b, there was no other scale available for comparison.

### 4.1 Description of calibration scales

#### 4.1.1 SF$_6$

**SIO-05**: SIO developed in the 1990s a preparation method using a bootstrap technique to make a suite of standards for halogenated compounds at pmol/mol levels (Prinn et al., 2000; Weiss et al., 1981). This technique relies on the preparation of a first primary standard for $CO_2$ at ppm levels (work of C. D. Keeling at SIO). $CO_2$ is then used as bootstrap gas to prepare a second mixture containing $CO_2$ and $N_2O$ with a known prepared $N_2O/CO_2$ ratio, close to atmospheric conditions. The $CO_2/N_2O$ ratio is prepared gravimetrically, by filling pure $CO_2$ and $N_2O$ in individual glass vials, flame-sealing the vials, weighing them, and mixing their content into an aliquot. In this second standard the $N_2O$ molar frac-

tion is assigned using this known, gravimetrically prepared $N_2O/CO_2$ ratio, and a $CO_2$ molar fraction calibration vs the $CO_2$ calibration scale. In a third step $N_2O$ is used as bootstrap gas to produce a new mixture containing $N_2O$ and halogenated gases, again with ratios close to ambient conditions. Knowing these halogen/$N_2O$ ratios and by measuring the $N_2O$ molar fraction against the $N_2O$ primary standard, each halogen molar fraction is determined. This preparation system therefore combines gravimetric preparation of ratios and measurement of one of them vs another suite of standards.

For comparison with the METAS-2017 calibration scale, we use so-called 'tertiary tanks' filled with real air at SIO and calibrated vs an SIO 'secondary standard', itself calibrated vs the suite of gravimetrically prepared, primary reference gas mixture defining the calibration scale for each compound. It is therefore necessary to take into account the uncertainty due to scale propagation, i.e. the measurement repeatability of the secondary vs the primary tank, and of the tertiary vs the secondary tank. The same principle applies to all other substances on a SIO calibration scale measured from a tertiary tank vs the METAS-2017 scale.

When preparing the primary calibration scales for halogenated gases, SIO did not assign an uncertainty following JCGM:2008, but still the internal consistency of the scale was precisely determined as being u = 0.4 % for SF$_6$

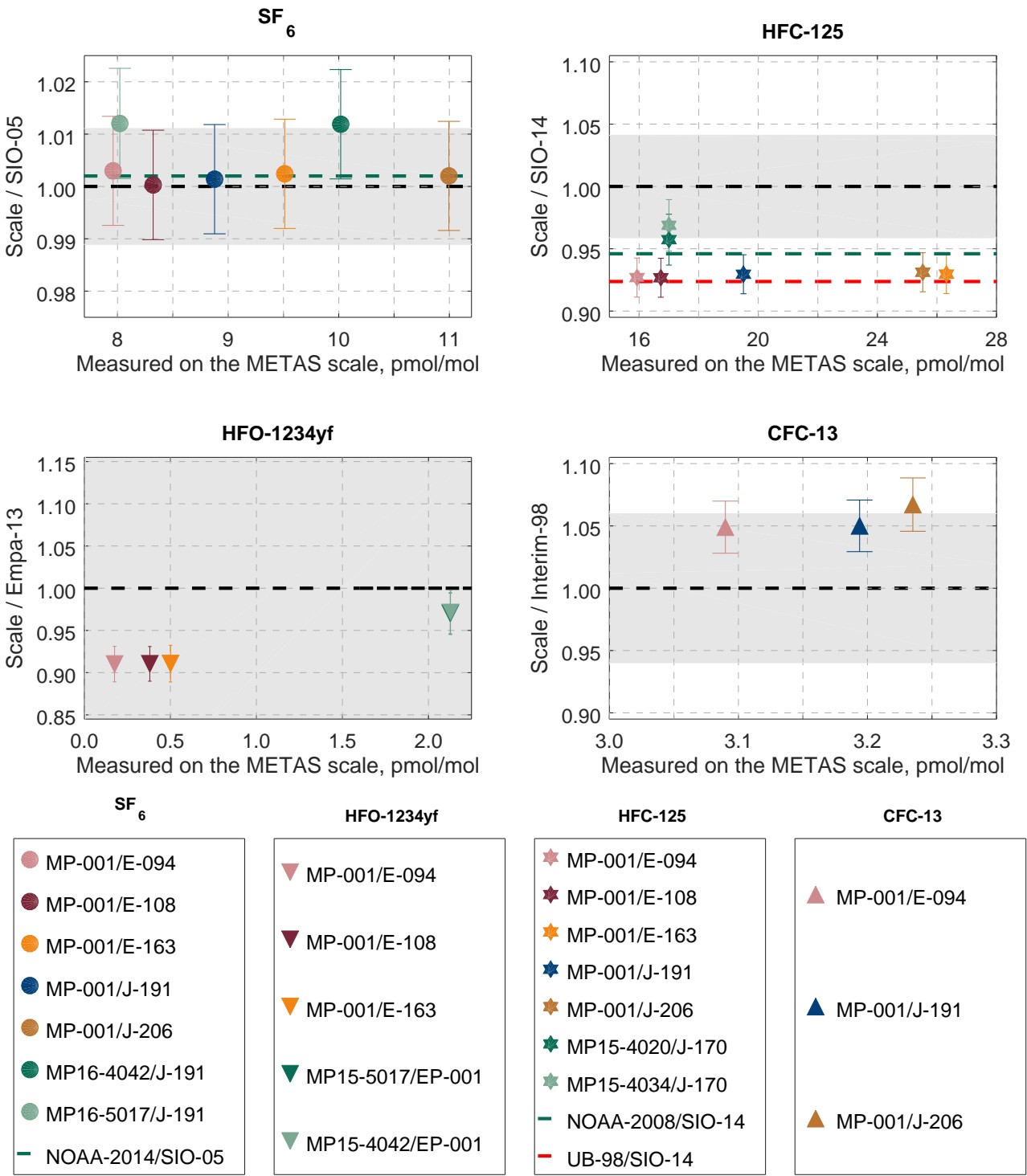

**Figure 6.** Comparison to existing calibration scales. Results are shown as ratios of values on the METAS scale divided by values on the historical scale. Results where both scales are in perfect agreement would line on the 1:1 line (dashed, black line). The grey area represents the uncertainty associated with the historical scale plus the scale transfer uncertainty (see description in Sect. 4.1). Markers represent measurement results of cylinder comparisons. Error bars on the markers take into account uncertainty of the METAS scales as well as the measurement reproducibility. Results on METAS and historical scales are in agreement within uncertainties as soon as the error bars touch the grey area. Additional dashed lines represent published conversion factors between SIO scales and other scales, i.e. NOAA (green) and University of Bristol (UB-98, red). An overview of results from this work and used conversion factors can also be found in Table 6.

**Table 6.** Scale comparison: individual cylinder measurement results and calculated average scale ratios. All measurements have been performed by Medusa-GC-MS at Empa Laboratories (see main text). We provide as well for documentation the known scale conversion factors used in this study: (a) Krummel et al. (2017), (b) Simmonds et al. (2017), (c) C.M. Harth and R.F. Weiss, pers. com., 2018.

| | $SF_6$ | HFC-125 | HFO-1234yf | CFC-13 |
|---|---|---|---|---|
| **Individual cylinder measurements** | | | | |
| MP15-4020/J-170 | - | 0.957 | - | - |
| MP15-4034/J-170 | - | 0.969 | - | - |
| MP15-5017/EP-001 | - | - | 0.970 | - |
| MP15-4042/EP-001 | - | - | 0.971 | - |
| MP16-4042/J-191 | 1.012 | - | - | - |
| MP16-5017/J-191 | 1.012 | - | - | - |
| MP-001/E-094 | 1.003 | 0.927 | 0.910 | 1.049 |
| MP-001/E-108 | 1.000 | 0.927 | 0.910 | - |
| MP-001/E-163 | 1.002 | 0.930 | 0.911 | - |
| MP-001/J-191 | 1.001 | 0.930 | - | 1.050 |
| MP-001/J-206 | 1.002 | 0.931 | - | 1.067 |
| **Calculated average scale ratios based on measurements** | | | | |
| METAS-2016/SIO-05 | 1.012 | - | - | - |
| METAS-2015/SIO-14 | - | 0.963 | - | - |
| METAS-2015/Empa-2013 | - | - | 0.971 | - |
| METAS-2017/SIO-05 | 1.002 | - | - | - |
| METAS-2017/SIO-14 | - | 0.929 | - | - |
| METAS-2017/Empa-2013 | - | - | 0.910 | - |
| METAS-2017/Interim-98 | - | - | - | 1.055 |
| METAS-2017/METAS-2015 | - | 0.964 | 0.938 | - |
| METAS-2017/METAS-2016 | 0.990 | - | - | - |
| **Known scale conversion factors used in this study** | | | | |
| NOAA-2014/SIO-05 | 1.002[a] | - | - | - |
| NOAA-2008/SIO-14 | - | 0.946[b] | - | - |
| UB-98/SIO-14 | - | 0.9237[c] | - | - |
| **Additionally calculated scale conversion factors** | | | | |
| METAS-2017/NOAA-2014 | 1.000 | - | - | - |
| METAS-2017/NOAA-2008 | - | 0.982 | - | - |

(R.F. Weiss, pers. com. Oct. 2017). Regular comparison with NOAA, comparing results at co-located monitoring stations (Rigby et al., 2010) or through cylinder exchanges (Hall et al., 2014), show agreement within 0.2 % or better for $SF_6$. NOAA recently determined the uncertainty of its $SF_6$ reference gas mixtures following JCGM:2008 as 0.062 pmol/mol ($k = 2$) for molar fractions in the range 7–10 pmol/mol, equivalent to 0.6 % to 0.9 % (Lim et al., 2017). The preparation method followed by NOAA has similarities with to the one developed at SIO, being based on static gravimetry as well. We therefore conservatively use an uncertainty for the SIO-05 assigned molar fraction of 1 % ($k = 2$).

**METAS-2016**: A set of two primary standards was prepared at METAS in 2016 (METAS-2016 calibration scale) to participate in an intercomparison for $SF_6$ in air at atmospheric molar fractions organised by the World Calibration Centre for $SF_6$ (Lee et al., 2017). These two standards were prepared using a similar method as for the METAS-2017 scale using permeation, dynamic dilution and cryo-filling of a nmol/mol molar fraction mixture containing $SF_6$ only in synthetic air (for more details see Lee et al., 2017). After homogenisation this mother mixture was dynamically diluted into two daughter mixtures at 8 and 10 pmol/mol, themselves transferred in cylinders by cryo-filling. The expanded uncertainty of the prepared standards is U = 1.3 %.

**NOAA-2014**: We use for comparison the known $SF_6$ conversion factor between the NOAA-2014 and SIO-05 calibration scales of NOAA-2014/SIO-05 = 1.002 $\pm$ 0.002 (Krummel et al., 2017), based on measurements at co-located stations and tank exchanges. This conversion factor is depicted by a green dashed line on Fig. 6.

### 4.1.2    HFC-125

**SIO-14**: We use the SIO calibration scale for HFC-125 prepared in 2014 following the same method as the SIO scale for $SF_6$. The estimated uncertainty for this scale is 4 % (Prinn et al., 2000).

**METAS-2015**: A first HFC-125 reference gas mixture was produced by METAS in 2015, with expanded uncertainty U = 2%. The preparation method consisted of a permeation step using MSB-Violetta, generating a mixture at approx. 85 nmol/mol, transferred to a SilcoNert-coated cylinder by cryo-filling. This mother mixture was then diluted into two daughter mixtures both at 17 pmol/mol, using dynamic dilution steps. Diluted daughter mixtures were then directly injected into the Medusa-GC-MS (Supplement, Sect. S4 and Fig. S1, S2 and S3).

**NOAA-2008**: We use the published conversion factor NOAA-2008/SIO-14 of $0.946 \pm 0.008$ (Supplement of Simmonds et al., 2017, p. 11). This factor was determined by comparing AGAGE continuous measurements by Medusa-GC-MS and NOAA flask samples taken from 3 co-located sites.

**UB-98**: Before using the SIO-14 scale for HFC-125 within AGAGE, a primary calibration scale prepared by University of Bristol was in use (UB-98, O'Doherty et al., 2004, 2009). The known conversion factor UB-98/SIO-14 is 0.9237 (C.M. Harth and R.F. Weiss, pers. com., 2018).

### 4.1.3    HFO-1234yf

**Empa-2013**: In 2013 Empa prepared a first calibration scale for a set of newly emitted compounds, including HFO-1234yf at $\approx 2$ pmol/mol, using volumetric dilution (Supplement of Vollmer et al., 2015). The molar fraction uncertainty is likely no more than $U \le 30$ %.

**METAS-2015**: Two reference gas mixtures for HFO-1234yf were produced in 2015 at 2 pmol/mol following the same method as for the METAS-2015 standards for HFC-125 (Supplement, Sect. S4). The resulting expanded uncertainty was U = 2.5%.

### 4.1.4    CFC-13

**Interim-98**: For CFC-13, a preliminary calibration scale was developed at the University of Bristol based on dilution of a high molar fraction reference gas mixture purchased from a commercial manufacturer (Linde Gas, hereafter Interim-98 scale, O'Doherty et al., 2004). This preliminary scale has been used within AGAGE until 2017. It would be very difficult to assign an uncertainty to this mixture, and it should be noted that the aim of this Interim-98 standard was to serve as intermediate anchor in order to be able to report CFC-13 internally within AGAGE. We tentatively assign an uncertainty of 5 %. The Interim-98 scale was transferred to the SIO suite of secondary standards by measurement comparisons, and

therefore the tertiary standards used at Empa have a CFC-13 assigned value on the Interim-98 scale. We provide comparison to this calibration scale to contribute to the documentation of the scale transfer from Interim-98 to METAS-2017 for CFC-13, rather than to realise a new intercomparison. Additional details on the scale transfer are given in Vollmer et al. (2018).

## 4.2    Results of comparisons

We express the results as ratio values, i.e. the value expressed on the METAS calibration scale divided by the value expressed on the historical calibration scale (SIO-05, SIO-14, Empa-2013 and Interim-98). A ratio of one would correspond to a perfect agreement between the two compared scales.

### 4.2.1    $SF_6$

The average ratio METAS-2017/SIO-05 using all available results is 1.002, i.e. the deviation from 1 is clearly within the ratio uncertainty (Fig. 6). This demonstrates that the two calibration scales are concordant with each other. $SF_6$ is also the substance for which the NOAA/SIO ratio is closest to 1 (NOAA-2014/SIO-05 ratio of 1.002). Combining these two ratios, one obtains a METAS-2017/NOAA-2014 ratio for $SF_6$ of 1.000. Such excellent agreements, in particular between standards produced by dynamic and static methods, can in addition to the reliability of both preparation methods be explained by the stability and non-reactivity (for instance low adsorptivity) of this substance.

### 4.2.2    HFC-125

The METAS-2017 calibration scale for HFC-125 is 7 % lower than the SIO-14 scale (METAS-2017/SIO-14 = 0.929). For HFC-125, the value assigned on the SIO-14 scale is not corrected for potential impurities in the pure HFC-125 substance used for the preparation (C.M. Harth, pers. com. 2017), while a 1 % correction is used for the METAS-2017 scale. Assuming HFC-125 sources were similar, and would both scales apply the same procedure, the disagreement would be reduced to 6 %. We plan for future reference gas mixture preparation to check the presence of substance impurities in permeators in a systematic way, to get a better estimate of the purity fraction as well as to quantify any potential cross-contamination, if any. For the METAS-2017 scale, we checked in particular the absence of HFC-132b as impurity in the HFC-125 permeator (see Section S5 in the Supplement).

Comparison of the METAS-2015 and SIO-14 calibration scales showed as well a METAS value lower than the SIO value, by 4 % (METAS-2015/SIO-14 = 0.963). The ratio NOAA-2008/SIO-14 for HFC-125 is 0.946, one of the largest discrepancies observed between SIO and NOAA for halogenated gases (pers. comm. P.B. Krummel and B.D.

Hall, Jan. 2018). Thus, comparing the SIO-14 calibration scale for HFC-125 to these three other scales (NOAA-08, METAS-2015, METAS-2017) points to a probable overestimation of the SIO-14 value (due e.g. to substance losses by adsorption) or that the values on the other calibration scales are underestimated (due to e.g. unaccounted for contamination). Cases of gas standards produced by dynamic methods yielding results lower than those produced by static method have already been observed several times with reactive substances prone to adsorption on surfaces, such as ammonia on stainless steel (van der Veen et al., 2010). This is due to the fact that when applying dynamic methods, potential losses by adsorption on surfaces can be canceled out when the generation process reaches equilibrium, after a sufficiently long stabilisation time. Interestingly, the METAS-2017 calibration scale is even lower than the METAS-2015 scale, by 3 %. Significant improvements in the generation process were made for the METAS-2017 scale to considerably minimise the total exposition to metal surfaces, compared to the METAS-2015 scale. This improvement is potentially the cause of the observed 3 % shift towards lower values.

### 4.2.3   HFO-1234yf

For HFO-1234yf, the two METAS calibration scales are lower than the Empa-2013 scale with in average METAS-2017/Empa-2013 = 0.910 and METAS-2015/Empa-2013 = 0.971. METAS-2017 is thus lower than METAS-2015 by 6 %. As with HFC-125, this latter ratio is significantly lower than 1. Within all halogenated substances studied here, this is the largest discrepancy observed between static (Empa-2013) and dynamic (METAS-2017) preparation methods, as well as the largest offset between different METAS scales. In addition, within the METAS-2017 suite of cylinders the largest offsets for outliers are also observed for HFO-1234yf (Fig.5, HFO-1234yf in MP-010 and MP-008 is ≈5.5 % too low). All these observations suggest that preparing primary reference gas mixtures for HFO-1234yf with U ≤ 2 % may require dynamic generation methods, with additional minimisation of contact with surfaces.

### 4.2.4   CFC-13

Three cylinders on the Interim-98 calibration scale for CFC-13 have been compared to cylinder MP-001 on the METAS-2017 scale. For documentation purposes, we report the average ratio METAS-2017/Interim-98 measured as 1.055 (Fig. 6). The comparison was extended to additional cylinders in Vollmer et al. (2018) to ensure a reliable scale transfer from Interim-98 to METAS-2017 within the AGAGE network.

## 5   Conclusions

We have developed a suite of primary, SI-traceable reference gas mixtures in 11 pressurised cylinders for $SF_6$, HFC-125, HFO-1234yf, HCFC-132b and CFC-13 in synthetic air, at atmospheric molar fractions. This suite constitutes the METAS-2017 primary calibration scales for these 5 halogenated compounds. This work therefore combines the advantages of SI-traceable reference gas mixture preparation with a primary calibration scale system for its use as anchor by a monitoring network. Such a combined system allows to maximise the compatibility (as defined by GAW) within the network while linking all reference values to the international system of units (SI) and assigning carefully estimated uncertainties following international guidelines (JCGM 100:2008).

Expanded uncertainties of the METAS-2017 calibration scale after verification ranges from 1 % to 2 % at a 95 % confidence interval. Such molar fractions at the pmol/mol level with associated expanded uncertainties of no more than 2 % clearly mark a step beyond the state of the art for dynamic methods. We have demonstrated the applicability of dynamic gravimetric generation methods coupled to cryofilling in cylinders to prepare primary reference gas mixtures for halogenated compounds as low as 1 pmol/mol. For stable compounds for which static gravimetric methods are also applicable (e.g., $SF_6$), these latter methods perform better in terms of expanded uncertainties (e.g., Lim et al., 2017), but we emphasise that using a completely independent preparation method may always help to detect potential systematic biases affecting one method or the other. From a metrological point of view, this preparation exercise is therefore highly valuable, ensuring comparability and redundancy of prepared reference gas mixtures.

Comparison of the METAS-2017 calibration scale for $SF_6$ with the scale prepared by SIO (SIO-05) leads to a conversion factor METAS-2017/SIO-05 of 1.002, illustrating the concordance of the two scales within uncertainties. An indirect comparison with the NOAA calibration scale also yields agreeing results (METAS-2017/NOAA-2014 = 1.000). The excellent concordance obtained for $SF_6$ gives confidence in the reliability of the presented dynamic-gravimetric method to prepare standards for other, more reactive compounds, e.g. HFC-125.

For HFC-125, known as more reactive than $SF_6$, the METAS-2017 calibration scale is measured as 7 % lower than SIO-14. In addition the METAS-2017 scale for HFO-1234yf is measured 9 % lower than Empa-2013. Such an offset towards lower values for standards prepared using dynamic generation methods by contrast to methods using static gravimetry or static volumetry has been previously observed for other reactive compounds such as ammonia. This underlines the risk of substance losses by e.g. adsorption on surfaces for HFC-125 and HFO-1234yf (and potentially other reactive substances). Dynamic generation meth-

ods and/or minimisation of contact on surfaces should therefore be favoured when preparing primary reference gas mixtures for such reactive substances.

*Data availability.* All data used to prepare the METAS-2017 suite and results of cylinder measurements within the suite are available in this article and its Supplement.

*Competing interests.* The authors declare no competing interest.

*Acknowledgements.* M.G. would like to thank members of the AGAGE community, and in particular Ray Weiss, for their interest in this work, which was a strong motivation to bring this work to the presented scientific level. Many thanks to the colleagues from METAS Gas Analysis laboratory, including H.P. Haerri now retired, and from other laboratories and technical departments, particularly the METAS workshop, who supported this work with their skills and their equipment. We thank collaborators of the HIGHGAS project for their valuable inputs: Fine Metrology Srls, SilcoTek GmbH, SAES Getters, Arbor Fluidtec AG, VICI Metronics AG. This project has been funded by METAS project AtmoChemECV and the EMRP project HIGHGAS. The EMRP is jointly funded by the EMRP participating countries within EURAMET and the European Union. Free softwares Inkscape and Octave were used for data analysis, diagrams and plots.

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
