# Peer review of "Dynamic-gravimetric preparation of metrologically traceable primary calibration standards for halogenated greenhouse gases"

_Atmospheric Measurement Techniques, 2018_

## Referee Comment (RC1) · Anonymous Referee #1 · 14 Mar 2018

This manuscript describes a dynamic dilution with cryogenic filling to produce pmol/mol reference materials in high-pressure cylinders. The method is well-described and the paper is well-written. The technique described is different to static dilution, which is commonly used to prepare low mole fraction reference standards for atmospheric measurement of trace gases that influence stratospheric ozone and climate. The authors provide comparisons to previous calibration scales, some of which are not well-developed. These data will improve our understanding of the atmospheric abundance of HCFC-132b, HFC-125, HFO-1234yf, and CFC-13. This work will help provide SI-traceability to current measurements, and the method could be useful for gases that are reactive or adsorb readily to dry surfaces.

I have relatively few comments and technical corrections.

Comments:

P5, L26: In equation (1), Vm is listed as the molar volume of the carrier gas (L/mol), which makes sense, but in Table 2, it is listed with units of g/mol. Is Vm correct in Table 2?

P5, L26: Is Vm calculated by assuming it is an ideal gas?

P7, L30: I'm having a hard time with equation (5). The units don't seem to work out. On page 8 you say that equation (5) can be simplified by removing  $q_V5$ , but this is not obvious. It seems that  $q_V5$  remains in the term  $x_{residual/(t_total*q_V5)}$ , unless an additional  $q_V5$  is missing from (5). Because of this and the confusion over Vm, I am unable to verify the calculations for SF6 in cylinder MP-001 using data from Table 2.

Table 2: The treatment of uncertainties seems reasonable, except for one minor component. You use the manufacturer's data for purity (99+%) and account for the uncertainty in the budget, which is acceptable. However, according to Vollmer at al. 2015 and references therein, HFC-125 is produced by hydrofluorination of perchloroethylene, with several intermediates, including HCFC-133a. Can you comment on the possibility that HFC-125 might contain HCFC-132b as an impurity? Does the purity uncertainty component for HCFC-132b need to be expanded to include this possibility?

Figure 6: I don't find this figure particularly helpful. It seems that the relevant information is in fig. 7 and Table S6.

Technical Corrections.

P1, L1: replace "withing" with "within"

P1, L15: Perhaps be more specific, "... traceable to the SI unit, amout of substance, ..."

P1, L23: Consider rephrasing: "Such a combined system supports maximizing com-
patibility ..."

P2, L9: (minor) Consider using mixing ratio or molar fraction instead of concentration

- P2, L13: Change "Kigali agreement" to "Kigali Amendment"
- P2, L16: Should probably spell out "Non-Article 5"

P2, L16: "bottom-up"

P2, L31: "detect gradients between"

P2, L32: "attribute" rather than "attributing"

P2, L29: Consider; "... while assessments of climate forcing and stratospheric ozone rely on observations of atmospheric composition".

P3, L11: No mention of what "compatibility target" is. Consider simplifying as "The calibration scale approach enables a high degree of consistency, but still requires ...."

P3, L12; Consider replacing "consists in" with "includes"

P5, L3: Consider re-phrasing. "The permeation rate depends exponentially on temperature; ..."

P12: I'm glad you included some possible reasons for some cylinders failing verification tests. Do you consider the possibility that some fraction of a component could be lost to the surface of the cylinder before the water is added? Maybe future experiments could be done in which the H2O is added earlier in the sequence?

P15, L14: change "apply for" to "applies to"

P15, L27: I think you have the NOAA/SIO ratio backwards. Rigby et al 2010 adjusted NOAA data by the factor 0.998, so that means that SIO/NOAA = 0.998, consequently NOAA/SIO would be 1.002. While the Rigby et al 2010 comparison is out of date (NOAA updated their SF6 scale from NOAA-2006 to NOAA-X2014), the ratio 1.002 is consistent, within uncertainties, with those calculated by P. Krummel
(http://www.wmo.int/pages/prog/arep/gaw/documents/GGMT2017\_T03\_Krummel.pdf).

Fig. 1: It would help if a box was drawn around the permeation chamber, similar to the box around MFM,MFC1, and MFC2.

Table 2: Is the entry for Permeation supposed to be ng/s instead of ng/min?

Vollmer, M. K., et al. (2015), Abrupt reversal in emissions and atmospheric abundanceaĂlof HCFC-133a (CF3CH2CI), Geo- p

---

## Referee Comment (RC2) · Anonymous Referee #2 · 16 Mar 2018

The manuscript by Guillevic et al. titled "Dynamic-gravimetric preparation of metrologically traceable primary calibration standards for halogenated greenhouse gases" describes the preparation of novel traceable gas standards containing SF6, HCFC-132b, HFC-125, HFO-1234yf, and CFC-13 in air by dynamic means using a permeation device. Many of these compounds are atmospherically important and no traceable reference standards are well established, therefore this is an important publication. The authors discuss the need and uses for these gravimetric standards and compare them to previous calibration scales.

On the whole, the article is well written, with a few minor typos that can easily be

amended. Please see below for my comments and feedback:

Minor changes: For coverage factors and confidence intervals k values should be in italics P1, line 1 remove 'g' from within P3, line 3 add an 'and' P3, line 12 replace 'in' with 'of' P3, line 13 remove end 's' from flasks P6, line 20 transferred 'into' P7, line 2 'checking' P10, line 1 why do you use k = 1?

Comments: In the introduction there is discussion about calibration scales and static point source measurements. It may be worth briefly commenting on atmospheric measurement and sampling of these compounds to highlight the challenges and needs for reference standards. In the introduction you may want to refer to WMO data quality objectives. Emphasise the impact of HFCs etc. on climate forcing and why the dynamically prepared reference standards are so important. One interesting paper is Velders et al. www.pnas.org_cgi_doi_10.1073_pnas.0902817106 Section 2.1 Please give the numeric calibration range. Section 2.4 Describe how equivalent cylinders (e.g. 3&4 or 2&9) were prepared in parallel. Section 3.1 State detection limits (lods). Regarding the stability of the permeation temperature, how critical is this? How can you tell if a sufficiently long stabilisation time has been reached to achieve equilibrium for the permeation device? Does the pressure of the cylinder have an influence? The authors suggest not, however the lower pressure cylinders seem to be more problematic, could this be attributed to wall effects? Surface reactions are mentioned for HFO-1234yf – was performance better in the treated cylinders? Why use stainless steel if dynamically produced standards often show lower values than statically prepared standards for reactive substances? Is your system not Silco treated? Figure 6 is unclear and I recommend removal or overhauling it.

I recommend that this manuscript be published once minor changes have been made. I look forward to seeing the final article.
* * *

---

## Author Response (AR2)

**Author's response**

Myriam Guillevic et al.

*Correspondence to:* Myriam Guillevic (myriam.guillevic@metas.ch)

**1  Response to comments from Reviewer 1**

**This manuscript describes a dynamic dilution with cryogenic filling to produce pmol/mol reference materials in high-pressure cylinders. The method is well-described and the paper is well-written. The technique described is different to static dilution, which is commonly used to prepare low mole fraction reference standards for atmospheric measurement**

5  **of trace gases that influence stratospheric ozone and climate. The authors provide comparisons to previous calibration scales, some of which are not well developed. These data will improve our understanding of the atmospheric abundance of HCFC-132b, HFC-125, HFO-1234yf, and CFC-13. This work will help provide SI traceability to current measurements, and the method could be useful for gases that are reactive or adsorb readily to dry surfaces. I have relatively few comments and technical corrections.**

10  The authors would like to express their gratitude to Reviewer 1 for his/her considerable effort to improve the quality of this manuscript with knowledgeable and concise comments and input. We hereafter provide point by point replies to them.

**P5, L26: In equation (1), Vm is listed as the molar volume of the carrier gas (L/mol), which makes sense, but in Table 2, it is listed with units of g/mol. Is Vm correct in Table 2?**

15  There is indeed a mistake in Table 2, the unit of $V_m$ should be L/mol, as in Equation (1). Table 2 is now corrected.

**P5, L26: Is Vm calculated by assuming it is an ideal gas?** We used NIST Tables of molar volumes (NIST Chemistry WebBook, <https://webbook.nist.gov>). The data are calculated for real gases, at 0°C and 1013.25 hPa. We added this information in the first occurrence of the quantity molar volume: 'Values are from the NIST Chemistry WebBook, assuming real gas'.

**P7, L30: Im having a hard time with equation (5). The units dont seem to work out. On page 8 you say that equation (5) can be simplified by removing $q_{V5}$, but this is not obvious. It seems that $q_{V5}$ remains in the term $x_{residual}/(t_{total} * q_{V5})$, unless an additional $q_{V5}$ is missing from (5). Because of this and the confusion over Vm, I am unable to verify the calculations for $SF_6$ in cylinder MP-001 using data from Table 2.**

25  There is indeed a mistake in Equation 5 as written in the manuscript, thank you for pointing this out. It should be written:

$$x_{prep,i,j} = \frac{x_{filled,i} \cdot q_{V5,j} \cdot \Delta t_{i,j} + x_{residual,i} \cdot q_{V5,j} \cdot (\Delta t_{total,j} - \Delta t_{i,j})}{\Delta t_{total,j} \cdot q_{V5,j}} \tag{1}$$

From the above correct equation, we can now simplify by removing the factor $q_{V5,j}$, and re-arrange to:

$$x_{prep,i,j} = x_{filled,i} \cdot \frac{\Delta t_{i,j}}{\Delta t_{total,j}} + x_{residual,i} \cdot \left(1 - \frac{\Delta t_{i,j}}{\Delta t_{total,j}}\right) \tag{2}$$

All calculations have been corrected and this does not change the results of the calculations, nor for the molar fractions neither for the ratios.

**Table 2: The treatment of uncertainties seems reasonable, except for one minor component. You use the manufacturers data for purity (99%) and account for the uncertainty in the budget, which is acceptable. However, according to Vollmer at al. 2015 and references therein, HFC-125 is produced by hydrofluorination of perchloroethylene, with several intermediates, including HCFC-133a. Can you comment on the possibility that HFC-125 might contain HCFC-132b as**

10 **an impurity? Does the purity uncertainty component for HCFC-132b need to be expanded to include this possibility?**

Thank you for pointing this out. We fully agree that cross-contamination originating from impurities present in permeators can potentially be an important issue. We plan to investigate this in a more systematic way for the next generation of reference mixtures, by performing measurements of impurities present in each permeator. We have modified Section 4.2.2 of the main text to mention this:

15 'We plan for future reference gas mixture preparation to check the presence of substance impurities in permeators in a systematic way, to get a better estimate of the purity fraction as well as to quantify any potential cross-contamination, if any. For the METAS-2017 scale, we checked in particular the absence of HFC-132b as impurity in the HFC-125 permeator (see Section S5 in Supplement).'

And we added the following section in the Supplement:

20 'Based on the findings from Vollmer et al. (2015) who identified several potential impurities in HFC-125 inherited from its production pathway, we have measured the presence of impurities in the permeator for HFC-125. We have done this test using the reference gas mixture for HFC-125 prepared at 85 nmol/mol (cylinder MP15-4020, see Fig. S1) as part of the METAS-2015 suite for HFC-125. These measurements have been performed by injecting 1 L of this mixture directly in the Medusa-GC-MS at Empa. The detection of the HFC-125 peak was disabled to avoid saturation of the detector. For HCFC-132b, there was no

25 chromatographic baseline excursion suggesting that the mole fraction was well below the detection limit (defined here as three times the noise level). As the amount of gas used for a measurement was 1 L, i.e. half the normal gas quantity of 2 L, we roughly estimate the detection limit for HCFC-132b as twice the detection limit for a normal measurement of 2 L, i.e. 0.03 pmol/mol (instead of 0.015 pmol/mol).

In addition, the measurements showed the presence of:

30     – CFC-115: 31 pmol/mol ($0.36 \cdot 10^{-3}$ mol per mol HFC-125)

    – HFC-143a: 51 pmol/mol ($0.6 \cdot 10^{-3}$ mol per mol HFC-125)

    – HFC-134a: 31 pmol/mol ($0.36 \cdot 10^{-3}$ mol per mol HFC-125)

    – HCFC-124: 11 pmol/mol ($0.13 \cdot 10^{-3}$ mol per mol HFC-125)

- SF$_6$: 0.2 pmol/mol with LOD = 0.03 pmol/mol ($2.5 \cdot 10^{-6}$ mol per mol HFC-125)

- HFO-1234yf: 0.06 pmol/mol with LOD = 0.02 pmol/mol, although we are not sure if this is a small impurity introduced potentially by contamination from a regulator

- CFC-13: no baseline disruption (LOD = 0.14 pmol/mol).

In cylinder MP-001 containing 32.027 pmol/mol HFC-125, the SF$_6$ impurities originating from the HFC-125 permeator correspond to a contribution of $2.5 \cdot 10^{-6} \cdot 32.027 = 0.00008$ pmol/mol SF$_6$. This can be neglected compared to the prepared 10.582 pmol/mol SF$_6$. The same conclusion applies to HFO-1234yf and to the other cylinders.'

**Figure 6: I dont find this figure particularly helpful. It seems that the relevant information is in fig. 7 and Table S6.** We agree and Figure 6 is now removed. Table S6 is therefore now included in the main text, as Table 7.

**P1, L1: replace "withing" with "within"** The text was corrected accordingly.

**P1, L15: Perhaps be more specific, "... traceable to the SI-units, amount of substance, ..."** The final unit we generate using the presented method contains indeed the mole, one of the SI-units, but to generate this unit we use as well for example flows, that are traceable to the SI-units second and meter. Also, the balance measures a mass, traceable to the unit kilogram, and the conversion to unit mole is realised using molar masses of molecules. Thus the final quantity, even being mole/mole, is traceable not only to the mole but to other SI units as well. We therefore prefer to keep the sentence as it is.

**P1, L23: Consider rephrasing: "Such a combined system supports maximizing compatibility ..."** Suggestion implemented.

**P2, L9: (minor) Consider using mixing ratio or molar fraction instead of concentration** Text modified accordingly, 'concentration' being substituted by 'molar fraction' for all occurrences.

**P2, L13: Change "Kigali agreement" to "Kigali Amendment"** Text modified accordingly.

**P2, L16: Should probably spell out "Non-Article 5"** Corrected.

**P2, L16: "bottom-up"** Corrected.

**P2, L31: "detect gradients between"** Corrected.

**P2, L32: "attribute" rather than "attributing"** Text modified accordingly.

**P2, L29: Consider; "... while assessments of climate forcing and stratospheric ozone rely on observations of atmospheric composition".** Text modified according to suggestion.

**P3, L11: No mention of what "compatibility target" is. Consider simplifying as "The calibration scale approach enables a high degree of consistency, but still requires ...."** Text modified according to suggestion.

**P3, L12; Consider replacing "consists in" with "includes"** Text modified according to suggestion.

**P5, L3: Consider re-phrasing. "The permeation rate depends exponentially on temperature; ..."** Text modified according to suggestion.

**P12: Im glad you included some possible reasons for some cylinders failing verification tests. Do you consider the possibility that some fraction of a component could be lost to the surface of the cylinder before the water is added? Maybe future experiments could be done in which the H2O is added earlier in the sequence?**

Indeed, there is always the possibility that some compounds are partially lost to the surface of the cylinder, and adding water earlier in the sequence of fillings would be a judicious approach. In this study, the strategy we applied to evidence potential losses on cylinder surfaces was to use cylinders of different volumes, filled at different pressures, with different coating materials. In the literature, tests of adding water after filling cylinders with gas mixtures for $CCl_4$ showed a desorption of $CCl_4$, suggesting that adding water afterwards may be acceptable in certain cases (Yokohata et al., 1985).

Besides, for cylinders which failed the verification test and which showed anomalously low levels of e.g. HFO-1234yf, we still don't know if the loss occurred due to adsorption in the cylinder or previously, on other stainless steel surfaced of the preparation system. This is a point we plan to investigate in the future by comparing molar fraction measured in the gas coming out of cylinders to molar fraction measured in the gas stream coming directly from the second dilution step of the magnetic suspension balance (MSB). This system being physically only a few meters apart from a new APRECON-GC-MS installed at METAS, it will be possible to have the MSB system connect to one inlet port of the APRECON-GC-MS, a cylinder filled with this same mixture to another port, and compare if the two molar fractions are identical, within measurement uncertainty. This was not done in this paper as the APRECON-GC-MS system was not validated yet. We hope to include this in a future study.

We included a paragraph in Section 3.2.2 (Verification test and exclusion of outliers) of the manuscript to discuss this:

'Furthermore, we plan to investigate if the observed substance losses occurred by adsorption on cylinders walls or beforehand in the preparation system. To do so, a comparison of the molar fraction in the mixture exiting the magnetic suspension balance/dynamic dilution system with the same mixture filled in cylinders will be performed. The recent installation of a measurement system for halogenated gases at METAS in the same laboratory makes this possible. If the adsorption indeed occurs in the cylinders, it will be tested if adding water vapour earlier in the sequence of fillings may help to limit this adsorption.'

**P15, L14: change "apply for" to "applies to"** Text corrected.

5    **P15, L27: I think you have the NOAA/SIO ratio backwards. Rigby et al 2010 adjusted NOAA data by the factor 0.998, so that means that SIO/NOAA = 0.998, consequently NOAA/SIO would be 1.002. While the Rigby et al 2010 comparison is out of date (NOAA updated their SF6 scale from NOAA-2006 to NOAA-X2014), the ratio 1.002 is consistent, within uncertainties, with those calculated by P. Krummel (http://www.wmo.int/pages/prog/arep/gaw/documents/GGMT2017_T03_Krummel.pdf).**

10    Thank you for this precise observation. We now use the ratio of 1.002 obtained by calculating the weighted mean of comparison ratios as presented by P. Krummel et al. (GGMT, 2017). Figure 7 was updated accordingly.

We also realised that on Figure 7, we show the known factor UB98/SIO for HFC-125 but it is not mentioned in the main text. We now added a small section:

**UB-98**: Before using the SIO-14 scale for HFC-125 within AGAGE, a primary calibration scale prepared by University of

15 Bristol was in use (UB-98 O'Doherty et al., 2004, 2009). The known conversion factor UB-98/SIO-14 is 0.9237 (C.M. Harth and R.F. Weiss, pers. com., 2018).

**Fig. 1: It would help if a box was drawn around the permeation chamber, similar to the box around MFM,MFC1, and MFC2.** Figure 1 is now modified according to suggestion. The permeation chamber is now drawn by an orange box. Addi-

20 tionally another box is drawn around the ensemble 'magnetic suspension balance', made of the balance plate and its associated electromagnet, the permeator hooked to the permanent magnet, both in the permeation chamber.

[Figure]

**Figure 1.** Schematic of the dynamic-gravimetric preparation method. MFM: thermal mass flow meter. MFC: thermal mass flow controller. PrC: pressure controller. bPrR: back (downstream) pressure regulator. V1 and V2: pneumatic valves.

**Table 2: Is the entry for Permeation supposed to be ng/s instead of ng/min?** The given value of $1654.77$ for the perme-ation rate is correctly assigned a unit of ng/min. However there is indeed a mistake in Table 2, because the time values $t1_{SF_6}$ and $t2_{SF_6}$ should be in minutes, not in seconds. Table 2 is now corrected accordingly. Table S1 contained the same unit mistake and is now corrected as well. Note that all calculations are still valid, and the units in Figure 2 are correct. This mistake likely comes from the fact that $\Delta t_{i,j}$ are reported in seconds.

**2  Response to comments from Reviewer 2**

**The manuscript by Guillevic et al. titled "Dynamic-gravimetric preparation of metrologically traceable primary cali-bration standards for halogenated greenhouse gases" describes the preparation of novel traceable gas standards con-taining SF$_6$, HCFC-132b, HFC-125, HFO-1234yf, and CFC-13 in air by dynamic means using a permeation device. Many of these compounds are atmospherically important and no traceable reference standards are well established, therefore this is an important publication. The authors discuss the need and uses for these gravimetric standards and compare them to previous calibration scales. On the whole, the article is well written, with a few minor typos that can easily be amended. Please see below for my comments and feedback.**

We thank the reviewer for his/her time and constructive comments. Please find below a point-by-point answer to each com-ment.

**Minor changes: For coverage factors and confidence intervals k values should be in italics.**

The text was corrected accordingly.

**P1, line 1 remove 'g' from within.**

Text corrected.

**P3, line 3 add an 'and'.**

Text modified according to suggestion.

**P3, line 12 replace 'in' with 'of'.**

Test modified, also according to suggestion by R1: 'This QA/QC procedure includes regular intercomparisons [...]'.

**P3, line 13 remove end 's' from flasks.**

Text corrected accordingly.

**P6, line 20 transferred 'into'.**

Text corrected accordingly.

**P7, line 2 'checking'.**

Text corrected accordingly.

**P10, line 1 why do you use $k = 1$?**

We use $k = 1$ for $u_{R_{prep,i,j}}$ because this standard uncertainty is then used in the bivariate weighted linear fit, together with $u_{R_{meas,i,j}}$ to find the parameters $a_i$ and $b_i$. The equations are detailed in the Supplement, Section S3.

10 **Comments: In the introduction there is discussion about calibration scales and static point source measurements. It may be worth briefly commenting on atmospheric measurement and sampling of these compounds to highlight the challenges and needs for reference standards.**

Thank you for this remark. Also in agreement with R1, we have now completed the third paragraph of the introduction, introducing atmospheric measurement and sampling:

15 'Atmospheric measurements of halogenated compounds are currently provided by several networks such as AGAGE (Advanced Global Atmospheric Gases Experiment), NOAA (National Oceanic and Atmospheric Administration), and GAW (Global Atmospheric Watch). Such measurements, used to precisely estimate atmospheric molar fraction of these halogenated substance together with associated trends, are crucial to understand and predict the evolution of stratospheric ozone and estimate their radiative forcing thereby refining future climatic projections. Furthermore based on these measurements and using

20 atmospheric transport modeling, emissions can be quantified ('top-down' estimation, e.g., Prinn et al., 2000; Rigby et al., 2010; Brunner et al., 2017). The comparison of top-down reconstructions with bottom-up inventories shows agreement for some gases but also discrepancies that can be considerable for others (Weiss and Prinn, 2011; Lunt et al., 2015; Hu et al., 2016; Simmonds et al., 2016; Sherry et al., 2017). The top-down approach thus is a complementing and independent way to review production/consumption/emission inventories and compliance with reduction targets, while assessments of climate forcing and

25 stratospheric ozone rely on observations of atmospheric composition'.

**In the introduction you may want to refer to WMO data quality objectives.**

Thank you for this comment. We did not mention WMO data quality objectives in this article for several reasons. First, WMO is only one of the several networks/groups measuring halocarbons (e.g., AGAGE, University of California, Goethe Universität

30 Frankfurt). To our knowledge, there are set DQO for $SF_6$ within WMO (Compatibility goal, 1 sigma: 0.02 pmol/mol and extended compatibility goal, 1 sigma: 0.05 pmol/mol), but not for the other halocarbons (see for example GAW Report No. 229, 18th WMO/IAEA Meeting on Carbon Dioxide, Other Greenhouse Gases and Related Tracers Measurement Techniques (GGMT-2015), Table 1 p. 3). The WMO DQO for $SF_6$ applies to the data produce within the WMO network, not to other networks, which may have different DQOs or different methods to measure data quality. As our suite of standards is meant to

35 be available to any group/network asking for it, we decided not to mention requirements of one specific network.

Second, the link between DQO as defined by WMO and expanded uncertainty of a produced reference gas mixture is not straightforward. It strongly depends on the method used to disseminate the primary standard to the end-users. Actually, within the scale system used by networks such as WMO for $SF_6$ or AGAGE for halocarbons, DQO becomes mostly relevant for the primary calibration scale propagation, not for its preparation. Within this framework, the best way to reach DQOs is to make the calibration chain between primary calibration scale and laboratory standard as short as possible as well as to improve measurement reproducibility. As the dissemination of the produced METAS-2017 primary calibration scales is beyond the scope of this paper, we think it is not very relevant to mention WMO DQOs for $SF_6$. The dissemination of the METAS-2017 scale for CFC-13 to the AGAGE network is described in Vollmer et al. (2018). Note that AGAGE does not use DQO as a measure of network data quality.

**Emphasise the impact of HFCs etc. on climate forcing and why the dynamically prepared reference standards are so important. One interesting paper is Velders et al. 2009.**

The impact of HFCs on climate forcing is mentioned in the first paragraph of the introduction, p. 2 l. 11-15. We thank Reviewer 2 for the suggested reference and have now modified the paragraph to include it: 'HFCs (hydrofluorocarbons) were introduced as replacement for CFCs and HCFCs. Their emissions, though not harmful to the ozone layer, are still increasing and contributing to global warming due to their high radiative forcing (Harris and Wuebbles, 2014, Velders et al., 2009). For this reason the recent Kigali Amendement (Oct. 2016) added these HFCs to the Montreal Protocol'.

The importance of dynamically prepared reference gas mixture for the mentioned gases was actually not well known before this study, therefore it is not mentioned in the introduction. However this finding is emphasised in Section 4.2, and especially when discussing the results for HFC-125 (Section 4.2.2) and HFO-1234yf (Section 4.2.3). It is mentioned again as the last sentence of the conclusion: 'Dynamic generation methods and/or minimisation of contact on surfaces should therefore be favoured when preparing primary reference gas mixtures for such reactive substances'.

**Section 2.1 Please give the numeric calibration range.**

We added to the first paragraph of Section 2.1 the following sentence: 'The resulting prepared molar fraction range covered by this suite varies between the five compounds, with a range of 0.9 – 1.5 pmol/mol for HFO-1234yf with the lowest molar fractions, to 26 – 42 pmol/mol for HFC-125 with the highest molar fraction (see details for each substance in Table 3)'.

**Section 2.4 Describe how equivalent cylinders (e.g. 3 & 4 or 2 & 9) were prepared in parallel.**

All cylinders were prepared in series, i.e. we use the same mixture exiting the MSB-dilution system to fill all cylinders, one after the other, as described in Section 2.4 p. 7: '[...] The cylinder valve is then manually closed, the cylinder disconnected and left standing vertically outside the building to warm up. A new cylinder is placed in the liquid nitrogen bath, connected to the filling system, and the filling procedure starts again'.

Equivalent cylinders are not filled in parallel but one after the other. For one given substance, equivalent cylinders are therefore not more correlated than any other couple of cylinders.

**Section 3.1 State detection limits (LODs).**

5    We have completed Section 3.1 as follow: 'Detection limits, defined here as three times the noise level, are 0.015 pmol/mol for $SF_6$, 0.02 pmol/mol for HFC-125, 0.01 pmol/mol for HFO-1234yf, 0.015 pmol/mol for HCFC-132b and 0.07 pmol/mol for CFC-13'.

**Regarding the stability of the permeation temperature, how critical is this? How can you tell if a sufficiently long**
10   **stabilisation time has been reached to achieve equilibrium for the permeation device?**

The sensitivity of the permeation rate to temperature is discussed in Section 2.5.1, p. 8, paragraph 'Permeation chamber temperature stability': 'Once carrier gas flow and pressure are kept constant, the permeation rate varies only with temperature. The stability of the permeation chamber temperature is 0.02 °C over 20 min ($k = 2$). Based on our experience measuring temperature sensitivity of permeation rate, this corresponds to approx. 0.1 % change in permeation rate'.

15   The estimation of the needed stabilisation time is explained in Section 2.2, p. 5, l. 22-24 : 'After inserting a permeation device, a stabilisation period is required mainly depending on chamber temperature, pressure and permeator membrane properties, before the mass loss becomes linear over time. This linear mass loss vs time is then determined for at least 8000 min to minimise the standard deviation of the measured mass loss due to balance noise.' We completed this information as follow: 'The time window $t_{2,i} - t_{1,i}$ during which the mass data are used is determined so that the residuals of the fit to the mass
20   loss over time are centered around zero and randomly distributed (see example for CFC-13 on Fig. 2 and Section S1 in the Supplement).'. We also modified the legend of Fig. 2 accordingly.

**Does the pressure of the cylinder have an influence? The authors suggest not, however the lower pressure cylinders**
**seem to be more problematic, could this be attributed to wall effects? Surface reactions are mentioned for HFO-1234yf**
25   **– was performance better in the treated cylinders? Why use stainless steel if dynamically produced standards often**
**show lower values than statically prepared standards for reactive substances?**

We discussed potential wall effects in Section 3.2.2, p. 12 l. 28-32: 'We however observe that most outliers are cases of substance loss (Fig. 5) and affect cylinders having the smallest total amount of gas filled and the highest surface/volume ratio (i.e. Essex cylinder 4.5 L, 24 bars). We would therefore in the future favour filling in cylinders of larger volume and pressure,
30   as well as a further automatising the cryo-filling process to limit human intervention as much as possible as well as to increase the safety of the procedure.'.

Following as well suggestion from R1, we added the following information: 'Furthermore, we plan to investigate if the observed substance losses occurred by adsorption on cylinders walls or beforehand in the preparation system. To do so, a comparison of the molar fraction in the mixture exiting the magnetic suspension balance/dynamic dilution system with the
35   same mixture filled in cylinders will be performed. The recent installation of a measurement system for halogenated gases at

METAS in the same laboratory makes this possible. If the adsorption indeed occurs in the cylinders, it will be tested if adding water vapour earlier in the sequence of fillings may help to limit this adsorption.'.

Yes, it seems performance was better in the cylinders passivated with SilcoNert2000 coating, as cylinders MP-003 to MP-006 show no outliers presenting a substance loss. All other cylinders are made of stainless steel passivated by electropolishing of extremely high quality (Essex Industries), with two different volumes and associated maximum pressures (see main text, Table 1). The larger type of cylinder has been used for decades by the AGAGE network and is moreover the container of choice for the Cape Grim Air Archive, showing excellent stability over time for most halogenated compounds (e.g., Prinn et al., 2000).

**Is your system not Silco treated?**

Thank you for this remark, indeed the preparation system is almost entirely SilcoNert2000-treated but this was not mentioned. The text is now modified in Section 2.3 according to this suggestion: 'Note that most metal surfaces in contact with the carrier gas and the produced gas mixture are passivated by applying SilcoNert2000 coating. This includes all metal tubing, all metal surfaces of the MFCs and MFM in contact with the gas, and most of the permeation chamber'.

**Figure 6 is unclear and I recommend removal or overhauling it.** We agree and Figure 6 is now removed.

**3   Additional corrections and changes**

The affiliation for Dr. Daiana Leuenberger has been updated.

In the Introduction p. 3 l. 4, the uncertainty range stated for NOAA primary standard preparation was for $SF_6$ only (0.6 to 0.9 %, Lim et al., 2017). However NOAA prepares primary reference gas mixtures for many other halocarbons, with usually higher uncertainties. The text is now modified to : '[...] 0.6 to <2 % for NOAA (Hall et al., 2007; Montzka et al., 2015; Lim et al., 2017)'.

It was not mentioned that molar fractions for halogenated compounds are calculated in dry synthetic air, i.e. the added water molar fraction is not included for the calculation of the halocarbon molar fraction. This is due to the fact that the Medusa-GC-MS used for analysis is equipped with a drying system. This is now mentioned in the text (Sect. 2.4), in Table 3 and Table 6.

There was a permutation mistake in Equations 1 and 2: the first denominator should be $(t_{2,i} - t_{1,i})$, otherwise the obtained mass loss is negative. This is now corrected. This has no influence on the results as permeation rate values were already correctly used as positive values in all calculations.

It is now mentioned in the abstract that the METAS-2017 scale for CFC-13 presented in this paper has been adopted by the AGAGE network: 'Finally, for CFC-13 the METAS-2017 primary calibration scale is 5% higher than the interim calibration scale (Interim-98) that was in use within the Advanced Global Atmospheric Gases Experiment (AGAGE) network before adopting the scale established in the present work'.

Figure 4 has been updated: it includes now results not only for $SF_6$ and CFC-13 but for all five halogenated compounds. To fit on one page the figure is therefore now shifted to landscape position.

Tables 4 and 5 in the discussion version have been merged into one Table to facilitate the layout for the final version (now Table 4).

The new Table 6 (before Table S6) was completed with a last section: now we give for documentation purpose the ratios METAS-2017/NOAA-2014 for $SF_6$ of 1.000 and METAS-2017/NOAA-2008 for HFC-125 of 0.982. These ratios are calculated using the measured METAS-2017/SIO-05 and METAS-2017/SIO-14 ratios for $SF_6$ and HFC-125 respectively (this study) as well as the know conversion factors between SIO and NOAA for these two compounds (see main text for references).

**References**

[revised manuscript text omitted]

25   ## S5   Measurement of impurities in the permeation device for HFC-125

Based on the findings from Vollmer et al. (2015) who identified several potential impurities in HFC-125 inherited from its production pathway, we have measured the presence of impurities in the permeator for HFC-125. We have done this test using

the reference gas mixture for HFC-125 prepared at 85 nmol/mol (cylinder MP15-4020, see Fig. S1) as part of the METAS-2015 suite for HFC-125. These measurements have been performed by injecting 1 L of this mixture directly in the Medusa-GC-MS at Empa. The detection of the HFC-125 peak was disabled to avoid saturation of the detector. For HCFC-132b, there was no chromatographic baseline excursion suggesting that the mole fraction was well below the detection limit (defined here as three times the noise level). As the amount of gas used for a measurement was 1 L, i.e. half the normal gas quantity of 2 L, we roughly estimate the detection limit for HCFC-132b as twice the detection limit for a normal measurement of 2 L, i.e. 0.03 pmol/mol (instead of 0.015 pmol/mol).

In addition, the measurements showed the presence of:

  – CFC-115: 31 pmol/mol ($0.36 \cdot 10^{-3}$ mol per mol HFC-125)

  – HFC-143a: 51 pmol/mol ($0.6 \cdot 10^{-3}$ mol per mol HFC-125)

  – HFC-134a: 31 pmol/mol ($0.36 \cdot 10^{-3}$ mol per mol HFC-125)

  – HCFC-124: 11 pmol/mol ($0.13 \cdot 10^{-3}$ mol per mol HFC-125)

  – $SF_6$: 0.2 pmol/mol with LOD = 0.03 pmol/mol ($2.5 \cdot 10^{-6}$ mol per mol HFC-125)

  – HFO-1234yf: 0.06 pmol/mol with LOD = 0.02 pmol/mol, although we are not sure if this is a small impurity introduced potentially by contamination from a regulator

  – CFC-13: no baseline disruption (LOD = 0.14 pmol/mol).

In cylinder MP-001 containing 32.027 pmol/mol HFC-125, the $SF_6$ impurities originating from the HFC-125 permeator correspond to a contribution of $2.5 \cdot 10^{-6} \cdot 32.027 = 0.00008$ pmol/mol $SF_6$. This can be neglected compared to the prepared 10.582 pmol/mol $SF_6$. The same conclusion applies to HFO-1234yf and to the other cylinders.

[Figure]

**Figure S1.** Overview of preparation steps for the 3 successive generations of scales prepared at METAS: METAS-2015, METAS-2016 and METAS-2017. The METAS-2017 scale is the most advanced scale in terms of minimisation of exposition to metal surfaces. ppb: nmol/mol. ppt: pmol/mol.

[Figure]

**Figure S2.** Preparation setup used to produce high concentration mixtures for HFC-125 and HFO-1234yf in cylinders.

[Figure]

**Figure S3.** Schematic of METAS '2-step-dilutor', a two-stage dynamic dilutor based on thermal mass flow controllers. This setup is used to dilute the high concentration cylinders for HFC-125, HFO-1234yf and $SF_6$ (see preparation overview in Fig. S1). The final mixtures are at pmol/mol levels.

**Table S1.** Mass loss rate determination for each permation device.

| | $m_1$, g | $m_2$, g | $t_1$, min | $t_2$, min | $q_m$, ng/min | $Stab_{balance}$, % (k = 1) |
|---|---|---|---|---|---|---|
| SF$_6$ | 28.44957646 | 28.43612094 | 5999.45 | 14130.80 | 1654.77 | 0.20 |
| HFC-125 | 28.56911448 | 28.56634188 | 1505.17 | 7128.32 | 493.07 | 0.40 |
| HFO-1234yf | 30.40242526 | 30.39984672 | 998.70 | 30205.77 | 88.28 | 0.41 |
| HCFC-132b | 29.46146154 | 29.46037128 | 89.83 | 10068.25 | 109.26 | 0.64 |
| CFC-13 | 30.09772063 | 30.09634635 | 5999.55 | 12319.95 | 217.44 | 0.57 |

**Table S2.** Flows used for the first and second dynamic dilution steps, in mL/min @ STP. MFM measures the total flow MFC1 + MFC2. MFC1 and MFC2 therefore do not need individual calibration.

| | MFC1 | MFC2 | MFM | MFC3 | MFC4 |
|---|---|---|---|---|---|
| **Set points** | | | | | |
| SF$_6$ | 167 | 4500 | - | 10 | 5000 |
| HFC-125 | 167 | 500 | - | 10 | 4000 |
| HFO-1234yf | 167 | 3500 | - | 10 | 4000 |
| HCFC-132b | 300 | 3500 | - | 10 | 4000 |
| CFC-13 | 300 | 3000 | - | 10 | 4000 |
| water vapour | 300 | 1200 | - | not used | not used |
| **Calibrated values** | | | | | |
| SF$_6$ | - | - | 4594.968 | 10.809 | 5115.463 |
| HFC-125 | - | - | 676.333 | 10.809 | 4109.936 |
| HFO-1234yf | - | - | 3691.690 | 10.809 | 4109.936 |
| HCFC-132b | - | - | 3817.535 | 10.809 | 4109.936 |
| CFC-13 | - | - | 3333.618 | 10.809 | 4109.936 |
| water vapour | - | - | not calibrated | not used | not used |

**Table S3.** Filling durations, in seconds.

| Cylinder | MP-001 | MP-002 | MP-003 | MP-004 | MP-005 | MP-006 | MP-007 | MP-008 | MP-009 | MP-010 | MP-011 |
|---|---|---|---|---|---|---|---|---|---|---|---|
| $SF_6$ | 4018 | 2040 | 1546 | 1546 | 1915 | 1178 | 1178 | 1166 | 1296 | 1060 | 1190 |
| HFC-125 | 4018 | 2040 | 1546 | 1546 | 1915 | 1178 | 1178 | 1166 | 1296 | 1060 | 1190 |
| HFO-1234yf | 4018 | 2040 | 1546 | 1546 | 1915 | 1178 | 1178 | 1166 | 1296 | 1060 | 0 |
| HCFC-132b | 4018 | 2040 | 1546 | 1546 | 1915 | 1178 | 1178 | 1166 | 1296 | 1060 | 1190 |
| CFC-13 | 4018 | 2040 | 1546 | 1546 | 1915 | 1178 | 1178 | 1166 | 1296 | 1060 | 1190 |
| water vapour | 24110 | 10200 | 8470 | 8470 | 6625 | 10310 | 7070 | 7130 | 6480 | 7660 | 8200 |

**Table S4.** Matrix gas impurity content: results of measurement by Medusa-GC-MS (molar fraction, pmol/mol). For the uncertainty budget we use a triangular distribution centered in this measured molar fraction, with the same value as half width of limit.

| | $SF_6$ | HFC-125 | HFO-1234yf | HCFC-132b | CFC-13 |
|---|---|---|---|---|---|
| Molar fraction, pmol/mol | 0.006 | 0.04 | 0.003 | 0.003 | 0.015 |
| Half width of limit, pmol/mol | 0.006 | 0.04 | 0.003 | 0.003 | 0.015 |

**Table S5.** Results of measured ratios without correction of analyser response. n is the number of replicate measurements. For cylinder MP-001, by default the ratio is set to 1, and we apply as standard deviation the average standard deviation over the set of cylinders, for each given substance. $u_{R_{meas}}$ is the standard deviation of the mean.

[revised manuscript text omitted]